# Linear and Nonlinear Characteristics of Long-Term NDVI Using Trend Analysis: A Case Study of Lancang-Mekong River Basin

**Xuzhen Zhong** [1,2,3,4,5], **Jie Li** [1,4,5], **Jinliang Wang** [1,3,4,5,*], **Jianpeng Zhang** [1,4,5], **Lanfang Liu** [1,4,5] and **Jun Ma** [1,6]

1   Faculty of Geography, Yunnan Normal University, Kunming 650500, China
2   School of Geography and Resource Science, Neijiang Normal University, Neijiang 641100, China
3   Southwest United Graduate School, Yunnan Normal University, Kunming 650500, China
4   Key Laboratory of Resources and Environmental Remote Sensing for Universities in Yunnan, Kunming 650500, China
5   Center for Geospatial Information Engineering and Technology of Yunnan Province, Kunming 650500, China
6   Department of Geology, Tomsk State University, Tomsk 634050, Russia
*   Correspondence: jlwang@ynnu.edu.cn; Tel.: +86-871-65941198

**Abstract:** Vegetation is the main body of the terrestrial ecosystem and is a significant indicator of environmental changes in the regional ecosystem. As an essential link connecting South Asia and Southeast Asia, the Lancang-Mekong River Basin(LMRB) can provide essential data support and a decision-making basis for the assessment of terrestrial ecosystem environmental changes and the research and management of hydrology and water resources in the basin by monitoring changes in its vegetation cover. This study takes the Lancang-Mekong River Basin as the study area, and employs the Sen slope estimation, Mann–Kendall test, and Hurst exponent based on the MODIS NDVI data from 2000 to 2021 to study the spatial and temporal evolution trend and future sustainability of its NDVI. Besides, the nonlinear characteristics such as mutation type and mutation year are detected and analyzed using the BFAST01 method. Results demonstrated that: (1) In the past 22 years, the NDVI of the Lancang-Mekong River Basin generally exhibited a fluctuating upward trend, and the NDVI value in 2021 was the largest, which was 0.825, showing an increase of 4.29% compared with 2000. However, the increase rate was different: China has the most considerable NDVI growth rate of 7.25%, followed by Thailand with an increase of 7.21%, Myanmar and Laos as the third, while Cambodia and Vietnam have relatively stable vegetation changes. The overall performance of NDVI is high in the south and low in the north, and is dominated by high and relatively high vegetation coverage, of which the area with vegetation coverage exceeding 0.8 accounts for 62%. (2) The Sen-MK trend showed that from 2000 to 2021, the area where the vegetation coverage in the basin showed a trend of increase and decrease accounted for 66.59% and 18.88%, respectively. The Hurst exponent indicated that the areas where NDVI will continue to increase, decrease, and remain unchanged in the future account for 60.14%, 25.29%, and 14.53%, respectively, and the future development trend of NDVI is uncertain, accounting for 0.04%. Thus, more attention should be paid to areas with a descending future development trend. (3) BFAST01 detected eight NDVI mutation types in the Lancang-Mekong River Basin over the past 22 years. The mutations mainly occurred in 2002–2018, while 2002–2004 and 2014–2018 were the most frequent periods of breakpoints. The mutation type of "interruption: increase with negative break" was changed the most during this period, which accounts for 36.54%, and the smallest was "monotonic decrease (with negative break)", which only accounts for 0.65%. This research demonstrates that combining the conventional trend analysis method with the BFAST mutation test can more accurately analyze the spatiotemporal variation and nonlinear mutation of NDVI, thus providing a scientific reference to develop ecological environment-related work.

**Keywords:** NDVI; spatial-temporal pattern; mutation detection; Hurst exponent; BFAST01; Lancang-Mekong River Basin

## 1. Introduction

In November 2014, the "Lancang-Mekong Cooperation" mechanism was proposed at the 17th China-ASEAN Leaders' Meeting. In March 2016, the Lancang-Mekong cooperation process was fully launched. Since June 2021, a series of statements and initiatives have been adopted, such as "Joint Statement on Strengthening Cooperation in the Sustainable Development of Lancang-Mekong Countries" and "Initiatives on Deepening Local Cooperation in Lancang-Mekong Countries" (http://www.lmcchina.org/index.html, accessed on 1 July 2022). This means that under the impetus of the "Lancang-Mekong Cooperation", the "Lancang-Mekong Economic Development Belt" has developed rapidly, significantly promoting the region's economic cooperation and development. However, some studies have shown that economic growth, human activities, and climate change will change and degrade the vegetation cover [1,2]. Since 1960, the Lancang-Mekong River Basin has encountered many challenges from climate change and rapid socio-economic development from human demand and the growth of urban and rural areas [3,4], which has further changed or even destroyed the ecosystems in countries along the Lancang-Mekong Economic Circle because the average basin-wide temperature increased by approximately 0.79 °C [3]. Vegetation, as an essential feature of surface habitat conditions, is not only a vital link connecting the atmosphere, soil, hydrosphere, and biosphere but also an essential indicator of global climate and ecosystem variations and crucial for the global carbon cycle [5–8]. It has become a key source of information for studying large-scale environmental changes [9]. Besides, normalized difference vegetation index (NDVI) changes acquired from remote sensing images are crucial for vegetation changes because they represent variability in the photosynthetic capacity of the vegetation [10], the higher the index value, the better the vegetation growth condition [11]. However, there is a lack of study on vegetation coverage in the LMRB globally, which is insufficient to provide sufficient support for vegetation protection and green economy construction in the region. Therefore, studying the NDVI dynamic changes and nonlinear mutation characteristics of the watershed plays a vital role in discovering and improving the vegetation coverage and problems in the watershed and ensuring the sustainable development of the economic environment.

Over the past several decades, the research on the dynamic change of vegetation has achieved fruitful results. From the research area and scale, there were global scales [12,13], national scales [14,15], regional scales [16,17], and basin scales [18,19]. From the research content, there were long-term NDVI temporal and spatial evolution and future sustainability analysis [20,21], the detection of driving factors [20], and the correlation analysis of NDVI, terrain, climate, and other elements [17,22], as well as mutation analysis of vegetation NDVI [23]. The commonly used research methods mainly include the linear regression analysis method [24,25], correlation analysis method [17,21,26], trend analysis method combining Theil–Sen slope and Mann–Kendall test [7,27,28], and Hurst exponent method [21,29]. Typical studies using these methods are as follows: Zhang et al. detected the long-term trends and abrupt changes of urbanized area by the Theil–Sen estimator and Mann–Kendall test [30], Yan et al. employed the Mann–Kendall test to test the significance of LAI [31], Tehrani et al. used the Mann–Kendall test to perform a correlation analysis of two months during the epidemic of COVID-19 [32], etc. Although the above methods are relatively classic and mature, and the research results provide a good reference for linear features such as improvement or degradation of NDVI trends, they lack the mutation detection of NDVI time series and are more expressive of a monotonic trend situation [23]. However, the gradual or abrupt characteristics of the NDVI trend should be considered when analyzing its spatiotemporal evolution in large-scale, long-term series. Some scholars have employed the BFAST (breaks for additive seasonal and trend) mutation detection method to study the nonlinear characteristics of vegetation NDVI. The BFAST method is a nonlinear trend analysis method with seasonality trend plus breakpoint detection [9], which is usually employed to detect the abrupt trend change in long-time series. It can divide the total trend into several segments, while its variants, BFAST01 (that is, defining at most one mutation point), can classify the trend of long-term vegetation NDVI, that

is, divide the trend into different types according to the breakpoint, which is very useful for comprehensively understanding and grasping of the dynamic change characteristics of vegetation cover. However, the application of the BFAST method in the ecological environment is relatively rare.

In summary, although many studies have been performed on the vegetation cover of the LMRB, there are still the following problems: First, the Lancang-Mekong River originates from the Qinghai Tibet Plateau and flows through the three provinces of Qinghai, Tibet, and Yunnan in China, as well as Myanmar, Laos, Thailand, Vietnam, and Cambodia, a total of six countries [33]. The basin is long and narrow in the north and south and connects China with South Asia and Southeast Asia. Its terrain is complex, and the altitude difference is huge [8]. Due to the region's particularity and data acquisition and processing difficulty, the current research on the basin NDVI mainly focuses on a specific local area. In contrast, the analysis of vegetation cover in the whole basin is relatively small. Secondly, regarding research content and methods, most of the research on NDVI in the LMRB is devoted to single-trend analysis methods, while no work employs various trend analysis methods on its temporal and spatial pattern, future trend, and the comparison between linear and nonlinear characteristics. Therefore, this study takes the whole watershed as the research object based on MODIS NDVI data from 2000 to 2021. Then, the Sen slope estimation, Mann–Kendall significance test, and Hurst model were combined to analyze the spatial distribution characteristics and trend change of vegetation coverage in the past 22 years and its future sustainable state. The BFAST01 method was then utilized to detect and explore the nonlinear characteristics, such as mutation type and year, to comprehensively and deeply understand the evolution law of NDVI in the area from multiple perspectives. This provides a scientific basis for the study area's ecological environment protection and sustainable development.

## 2. Materials and Methods

### 2.1. Study Area

The Lancang-Mekong River (LMR) is the sixth largest river in the world. It originates from Jifu Mountain, Zaduo County, Yushu Tibetan Autonomous Prefecture, Qinghai Province, China. After it flows to Qamdo, it is called the Lancang River. Then, it continues southward to the exit of the Nanla River in Yunnan Province. After leaving the country, it is called the Mekong River, which is injected into the Pacific Ocean south of Ho Chi Minh City, Vietnam, and it flows through a total of six countries, including China, Myanmar, Thailand, Laos, Cambodia, and Viet Nam (Figure 1). The LMRB covers an area of about 810,000 km$^2$, with wide upstream and downstream rivers and narrow middle reaches. The topography in the basin is undulating, the terrain is complex, the north is high and the south is low, and it has diverse climate types. The basin can be divided into two parts: The upper LMRB comprises an area in China known as the Lancang River, comprising the Tibetan Plateau, the Three Rivers Area, and Lancang Basin in China and Myanmar. The lower LMRB comprises the area downstream of the border between China and Lao PDR, and it is made up of the Northern Highlands, the Khorat Plateau, the Tonle Sap Basin, and the Mekong Delta (https://www.mrcmekong.org/about/mekong-basin/geography/geographic-regions/, accessed on 5 July 2022).

The climate of the LMRB ranges from high-altitude continental and temperate in the upper basin to tropical monsoonal in the lower basin. In the upper LMRB, much of this basin is snow-covered in the wet season with sub-zero temperatures. During the dry season, average temperatures remain low at around 13 °C. The climate of the lower Mekong River Basin is classified as tropical monsoonal and is dominated by the Southwest Monsoon. Various vegetations in the basin indicate the apparent latitude and vertical zonality under the influence of altitude and climate change. The main vegetation types are subtropical and temperate evergreen coniferous forests, evergreen broad-leaved forests, deciduous broad-leaved forests, mixed forests, closed shrubs, subalpine bushes, alpine meadows, wetlands, swamp forests, and mangroves [34].

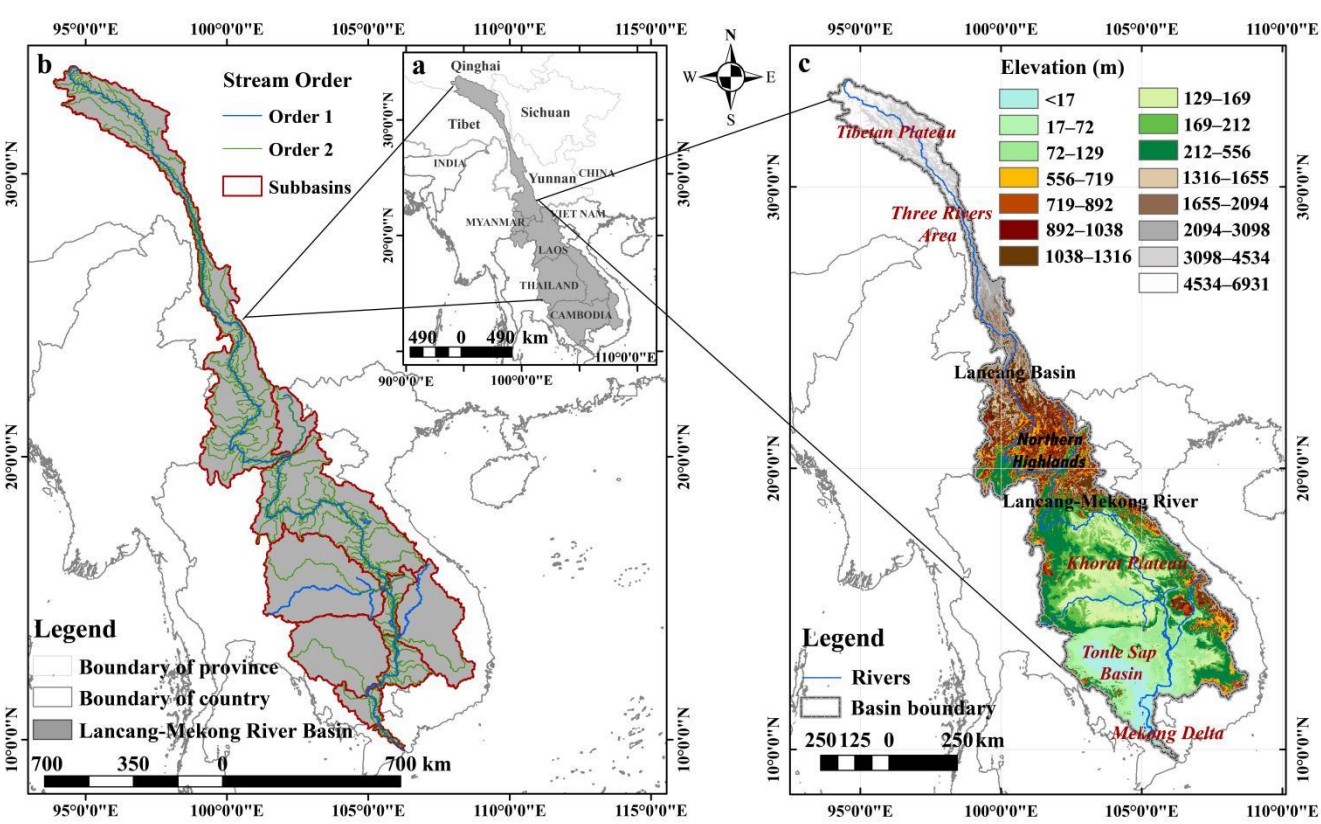

**Figure 1.** Lancang-Mekong River Basin. (**a**) Geographical location; (**b**) river network; (**c**) elevation.

## 2.2. Data Source and Preprocessing

This study employs MODIS NDVI data, DEM data, land-use data, and vector data of administrative divisions in the study area. The NDVI data were extracted from the MOD13Q1 version 6 product (Vegetation Indices 16-Day L3 Global 250 m) from 2000 to 2021, acquired from the USGS (https://lpdaac.usgs.gov/products/, accessed on 18 June 2022), and employed in the GEE platform (UIL: https://earthengine.google.com/, accessed on 18 June 2022) [7,35]. The MODIS NDVI data preprocessing in the study area, such as splicing, projection, and clipping, was performed through the GEE platform. The maximum value composite (MVC) method was then utilized to attain the monthly and annual maximum NDVI dataset for dynamic vegetation analyses [7]. The MVC method can remove most of the cloud and atmosphere effects. Its removal effect on cloud and bidirectional reflectance noise is obvious, with high computing efficiency, which is conducive to the production of vegetation index products, can provide users with important reference values of NDVI [36,37], and is one of the most widely used algorithms in NDVI data processing [38]. Many studies have used this algorithm [39,40] and some studies have mentioned that MVC images are highly correlated with the dynamics of green vegetation [41]. Then, the above monthly NDVI data were divided into four seasons of spring (March to May), summer (June to August), autumn (September to November), and winter (December to February of the next year) with reference to the seasonal classification standard of meteorology. Digital elevation model (DEM) data were collected from the global DEM (ASTER GDEM) with a spatial resolution of 30 m, which was derived from the Geospatial Data Cloud (http://www.gscloud.cn/, accessed on 18 June 2022). The data were mosaicked using ENVI5.1 software and clipped using ArcGIS 10.2 software. The land cover type data come from the European Space Agency (ESA) WorldCover 10 m 2020 product acquired from the ESA (https://esa-worldcover.org/en, accessed on 18 June 2022), and employed and clipped by the GEE platform. Other data, such as watershed scope, were acquired from the HydroSHEDS website (https://www.hydrosheds.org/, accessed on 9 June 2022), and the administrative division data of the study area come from 1 million

primary geographic data around the world (https://www.resdc.cn/data.aspx?DATAID=259, accessed on 1 June 2022). Table 1 lists the data sources.

**Table 1.** Source and description of research data.

| Name and Description | Spatial Resolution | Data Sources |
|---|---|---|
| MODIS NDVI data | 250 m | Google Earth Engine(https://earthengine.google.com/, accessed on 18 June 2022)Original data website: https://lpdaac.usgs.gov/products/ ( accessed on 18 June 2022) |
| DEM data | 30 m | Geospatial Data Cloud (http://www.gscloud.cn, accessed on 18 June 2022) |
| 2020 LULC Products | 10 m | Google Earth Engine(https://earthengine.google.com/, accessed on 18 June 2022)Original data website: https://esa-worldcover.org/en (accessed on 18 June 2022) |
| Watershed Boundary Data | 15 arc-second resolution | HydroSHEDS website (https://www.hydrosheds.org/, accessed on 9 June 2022) |
| Administrative division data | 1:1 million | Resource and Environment Science and Data Center of Chinese Academy of Sciences (http://www.resdc.cn, accessed on 1 June 2022) |

*2.3. Methods*

2.3.1. Sen's Slope Estimation and Mann–Kendall Significance Test

Sen slope estimation is a robust nonparametric statistical trend calculation method [7], and Mann–Kendall can be utilized to judge the significance of Sen slope estimation results [1]. In order to judge the trend of long-time series data, the Sen slope estimation and Mann–Kendall significance test can be combined [42]. The advantage of this combination is that it does not require the data to obey a particular distribution, is less affected by data outliers, and has strong robustness to data errors. Thus, it is suitable for analyzing the spatiotemporal variation of long-term NDVI [7,29,43]. Among them, the Sen slope estimation calculates the median of the slopes of $n(n-1)/2$ data combinations, and its calculation formula is:

$$S_{NDVI} = median\left(\frac{NDVI_j - NDVI_i}{j-i}\right), (2000 \leq i < j \leq 2021) \tag{1}$$

where $S_{NDVI}$ is utilized to represent the changing trend of NDVI, and $NDVI_i$ and $NDVI_j$ represent the values in years $i$ and $j$, respectively. If $S_{NDVI} > 0$, then the NDVI value has an upward trend in the study period; otherwise, if $S_{NDVI} < 0$, the NDVI value shows a downward trend in the study period. According to the relevant research results [44,45], with $-0.0005$ and $0.0005$ as the boundaries, the area between $-0.0005$ and $0.0005$ is divided into the no-change region. The area with values greater than or equal to $0.0005$ is divided into an increase region, and the area with values less than $-0.0005$ is divided into a decrease region.

The significance of the above trends will be further verified by the Mann–Kendall nonparametric statistical test, described as follows:

Set up a time series $\{NDVI_i\}, i = 2000, 2001, \cdots 2021$

$$Z = \begin{cases} \frac{S-1}{\sqrt{var(S)}} & (S > 0) \\ 0 & (S = 0) \\ \frac{S+1}{\sqrt{var(S)}} & (S < 0) \end{cases} \tag{2}$$

$$S = \sum_{i=1}^{n-1} \sum_{j=i+1}^{n} sgn(NDVI_i - NDVI_j) \tag{3}$$

$$sgn(NDVI_i - NDVI_j) \begin{cases} 1 \ (NDVI_i - NDVI_j) > 0 \\ 0 \ (NDVI_i - NDVI_j) = 0 \\ -1 \ (NDVI_i - NDVI_j) < 0 \end{cases} \tag{4}$$

$$var(S) = \frac{n(n-1)(2n+5)}{18} \tag{5}$$

where $NDVI_i$ and $NDVI_j$ have the same meaning as above; n$n$ represents the time series length; sgn is a symbolic function; and the value range of the statistic $Z$ is $(+\infty, -\infty)$, which can evaluate the significance of *NDVI*. At a given significance level $\alpha$, if $|Z| > Z_{1-\alpha/2}$, there are significant changes in the study sequence at $\alpha$ level. This research takes $\alpha = 0.05$, and the corresponding value is 1.960, to judge the significance of the trend of the NDVI time series at the confidence level of 0.05. The definition of significant change is ($|Z| \geq 1.96$), while the insignificant change is ($|Z| < 1.96$). This study couples the Sen slope estimation and Mann–Kendall significance test to obtain the Sen-MK trend and the specific division is shown in Table 2.

**Table 2.** Sen-MK trend type division.

| S$_{\mathbf{NDVI}}$ | Z Value | Sen-MK Trend Type |
|---|---|---|
| Increase region S$_{NDVI} \geq 0.0005$ | Significant Z $\geq 1.96$ | Significant increase |
| Increase region S$_{NDVI} \geq 0.0005$ | Insignificant $-1.96 < Z < 1.96$ | Weak increase |
| No change $-0.0005 < $S$_{NDVI} < 0.0005$ | Insignificant $-1.96 < Z < 1.96$ | No change |
| Decrease region S$_{NDVI} < 0.0005$ | Insignificant $-1.96 < Z < 1.96$ | Weak decrease |
| Decrease region S$_{NDVI} < 0.0005$ | Significant Z $< -1.96$ | Significant decrease |

2.3.2. Hurst Analysis

The Hurst exponent represents structure over asymptotically longer periods [46]. It is an effective method to quantitatively describe the long-range dependence of time series, and is a measure of a data series' "mild" or "wild" randomness [47,48]. It has to do with time series autocorrelations and the pace at which they drop as the lag between pairs of values grows longer [48]. It was established in hydrology for the purpose of calculating the optimum dam size [49]. In recent years, it has been widely utilized in future sustainability studies of vegetation cover changes [19,50]. The calculation process and principle are as follows:

(1)　Assume the time series $NDVI_t$, where $t = 1, 2, \ldots, n$.
(2)　Construct the mean sequence of $NDVI_t$:

$$\overline{NDVI_{(\tau)}} = \frac{1}{\tau} \sum_{t=1}^{\tau} NDVI_{(t)} \tag{6}$$

where $\tau$ is a positive integer greater than $t$.

(3)　Calculate the cumulative deviation $U_{(t,\tau)}$:

$$U_{(t,\tau)} = \sum_{t=1}^{\tau} \left( NDVI_{(t)} - \overline{NDVI}_{(\tau)} \right) \tag{7}$$

(4)　Calculate the range $R_{(\tau)}$

$$R_{(\tau)} = \max U_{(t,\tau)} - \min U_{(t,\tau)} (1 \leq t \leq \tau; \ \tau = 1, 2, 3 \cdots, n) \tag{8}$$

(5)　Calculate the standard deviation $S_\tau$:

$$S_\tau = \left[ \frac{1}{\tau} \sum_{t=1}^{\tau} \left( NDVI_{(t)} \right) - \overline{NDVI}_{(\tau)}^2 \right]^{\frac{1}{2}} \tag{9}$$

(6) Finally, calculate the Hurst exponent by the following formula:

$$\frac{R_\tau}{S_\tau} = (c\tau)^H \qquad (10)$$

where $c$ is the proportional parameter. Consider that there is a constant H satisfying $R(\tau)/S(\tau) \propto \tau^H$. This indicates a "Hurst effect" in the *NDVI* time series. H is the Hurst exponent value obtained by least squares fitting in the double logarithmic coordinate system.

This study determines the persistency of the *NDVI* sequence according to the H value. The H value is between 0 and 1. Referring to the relevant research results [7,24,29], the persistence characteristics of the H value are as follows: (1) If 0 < H < 0.5, then the *NDVI* time series has long-term dependence and the overall trend in the future contrary to the past, that is, the increasing trend in the past indicates a general decrease in the future and vice versa. This phenomenon is called anti-persistence, and the closer the value of H is to 0, the stronger the anti-persistence; (2) if H = 0.5, the *NDVI* time series is in an independent state and has no persistence; (3) if 0.5 < H < 1, the time series has long-term dependence, that is, it has persistence, indicating that the future change of *NDVI* will be compatible with the past change trend, and the increasing trend in the past will continue in the future, and vice versa, and the closer the value of H is to 1, the stronger persistence.

### 2.3.3. BFAST Mutation Test

BFAST is an iterative algorithm that utilizes piecewise linear trend and seasonal models to decompose time series into trend, seasonal, and residual components. It can detect abrupt changes in trend and seasonal components [51,52] and analyze different types of time series (e.g., Landsat, MODIS), and has been successfully adopted and validated in various ecosystems, such as in vegetation NDVI detection [53]. Mathematically, the trend and seasonal components of BFAST can be obtained by the following decomposition model:

$$Y_\mathrm{t} = T_t + S_t + e_t, ..t = 1, \ldots, n \qquad (11)$$

where $Y_\mathrm{t}$ is the value observed at time $t$, $T_t$ is the trend component, $S_t$ is the seasonal component, and $e_t$ is the residual component.

According to the BFAST principle [52,54], this study adopts the BFAST's improved model, called BFAST01. Compared with the BFAST model, the BFAST01 model considers both seasonal and trend models. It only detects the significant changes in time series while ignoring the minor structural changes to detect the most influential mutation in the trend to divide the trend into two segments instead of multiple smaller trends. That is to say, BFAST01 only detects 0 or 1 breakpoints [9,55]. This study employs the R environment to load the latest BFAST package (https://github.com/, accessed on 18 August 2022) from GitHub to complete all statistical analyses of BFAST01 [56]. The annual NDVI grid data of the LMRB from 2000 to 2021 are adopted for a total of 22 years to form a time series, and the BFAST01 program is employed for detection. The setting level is 0.05, the minimum breakpoint interval is 3, and the breakpoint for the detection methods is 5. Accordingly, the breakpoints in the NDVI time series of the basin are detected, and the trend significance, change magnitude, mutation type, and mutation year of the NDVI are extracted according to the method of priority detection of breakpoints.

In order to describe the change characteristics of NDVI before and after the trend mutation point detected by BFAST01 in a more intuitive and detailed manner, the pixel calculation results are classified and analyzed. Based on the studies of De Jong et al. [57] and Higginbottom et al. [9], this research divides the NDVI trend categories detected by BFAST01 into six categories. This situation is classified separately, considering no breakpoints in the NDVI time series. Finally, the trend types of NDVI changes are divided into eight categories (Table 3 and Figure 2) and four significance types, including both segments are significant (or no break and significant), only the first segment is significant,

only the second segment is significant, and both segments are insignificant (or no break and not significant). The validation of trend classification results used Google Earth time-series images.

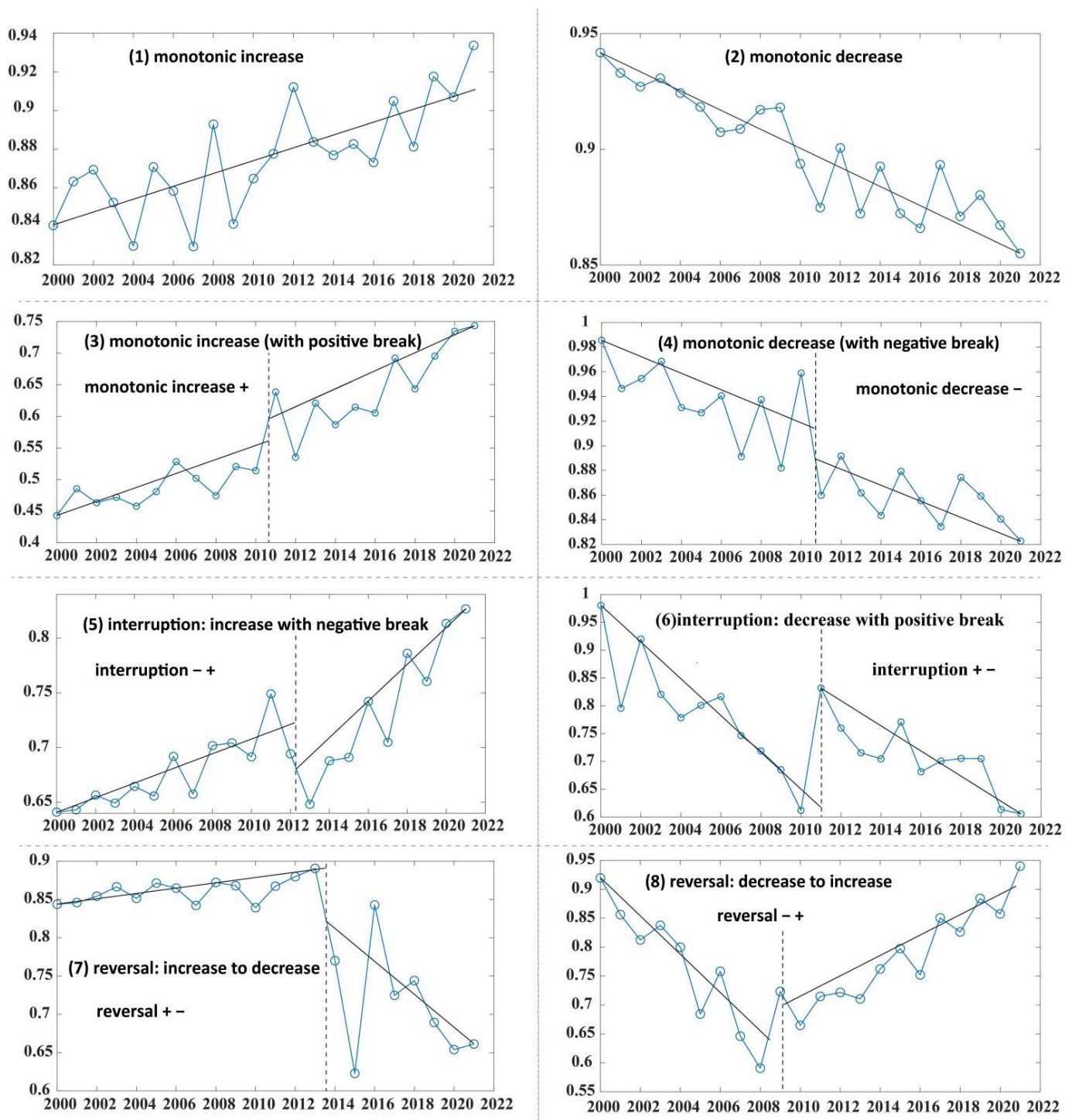

**Figure 2.** Schematic diagram of BFAST01 trend mutation types.

**Table 3.** Types and meanings of trend mutation in BFAST01.

| Type name | Meanings |
|---|---|
| (1) monotonic increase | No obvious mutation was detected, and the overall trend showed a monotonic increase |
| (2) monotonic decrease | No obvious mutation was detected, and the overall trend showed a monotonic decrease |
| (3) monotonic increase (with positive break) | One obvious mutation was detected, and the value at the breakpoint suddenly increased, and the overall trend showed a monotonic increase. Represented by the symbol "monotonic increase+". |
| (4) monotonic decrease (with negative break) | One obvious mutation was detected, and the value at the breakpoint suddenly decreased, and the overall trend showed a monotonic decrease. Represented by the symbol "monotonic decrease−". |
| (5) interruption: increase with negative break | One obvious mutation was detected, and the value at the breakpoint suddenly decreased, and the trend showed a significant increase with significant negative break followed by significant increase. Represented by the symbol "interruption−+". |
| (6) interruption: decrease with positive break | One obvious mutation was detected, and the value at the breakpoint suddenly increased, and the trend showed a significant decrease with significant positive break followed by significant decrease. Represented by the symbol "interruption+−". |
| (7) reversal: increase to decrease | One obvious mutation was detected, with a trend of switching from a significant increase to a significant decrease. Represented by the symbol " reversal+−". |
| (8) reversal: decrease to increase | One obvious mutation was detected, with a trend of switching from a significant decrease to a significant increase. Represented by the symbol " reversal−+". |

## 3. Results

### 3.1. Spatial and Temporal Pattern Analysis of NDVI in Lancang-Mekong River Basin

3.1.1. Time Change Analysis of NDVI from 2000 to 2021

The interannual variation line and growth rate graph of NDVI were made (Figure 3a) through the statistics of the average annual NDVI of the study area. As shown in Figure 3a, in the past 22 years, the NDVI of the LMRB had generally fluctuated and increased, with the average NDVI value of 0.804 for many years, and a maximum NDVI value of 0.825 in 2021, indicating an increase of 4.29% compared with 2000. However, the increasing rate and range were different: China has the largest NDVI growth rate of 7.25%, followed by Thailand with a 7.21% increase, and Myanmar and Laos third. In contrast, Cambodia and Vietnam have relatively stable vegetation changes. According to the line chart of annual mean changes (Figure 3a), NDVI exhibited a significant downward trend in 2004 and 2018, and the lowest NDVI value in 2004 was 0.7892. In order to further analyze the variation trend of NDVI in each season of the year, the multi-year average values of NDVI in spring (March to May), summer (June to August), autumn (September to November), and winter (December to February of the next year) of the basin from 2000 to 2021 were studied and calculated. The interannual change trend chart of NDVI in each season of the basin has been made (Figure 3b–f). The results indicated that the seasonal NDVI distribution characteristics in the study area were apparent, and the changing range of NDVI in each season was relatively stable over 22 years. Among these, the NDVI values in spring and winter were close and fluctuated slightly. The NDVI values in summer and autumn were close to each other, with slight fluctuations and an upward trend. The lowest NDVI values in spring and winter appeared in 2005 and 2004, respectively, and the NDVI values in summer and autumn decreased around 2018, coinciding with the turning point of the annual NDVI.

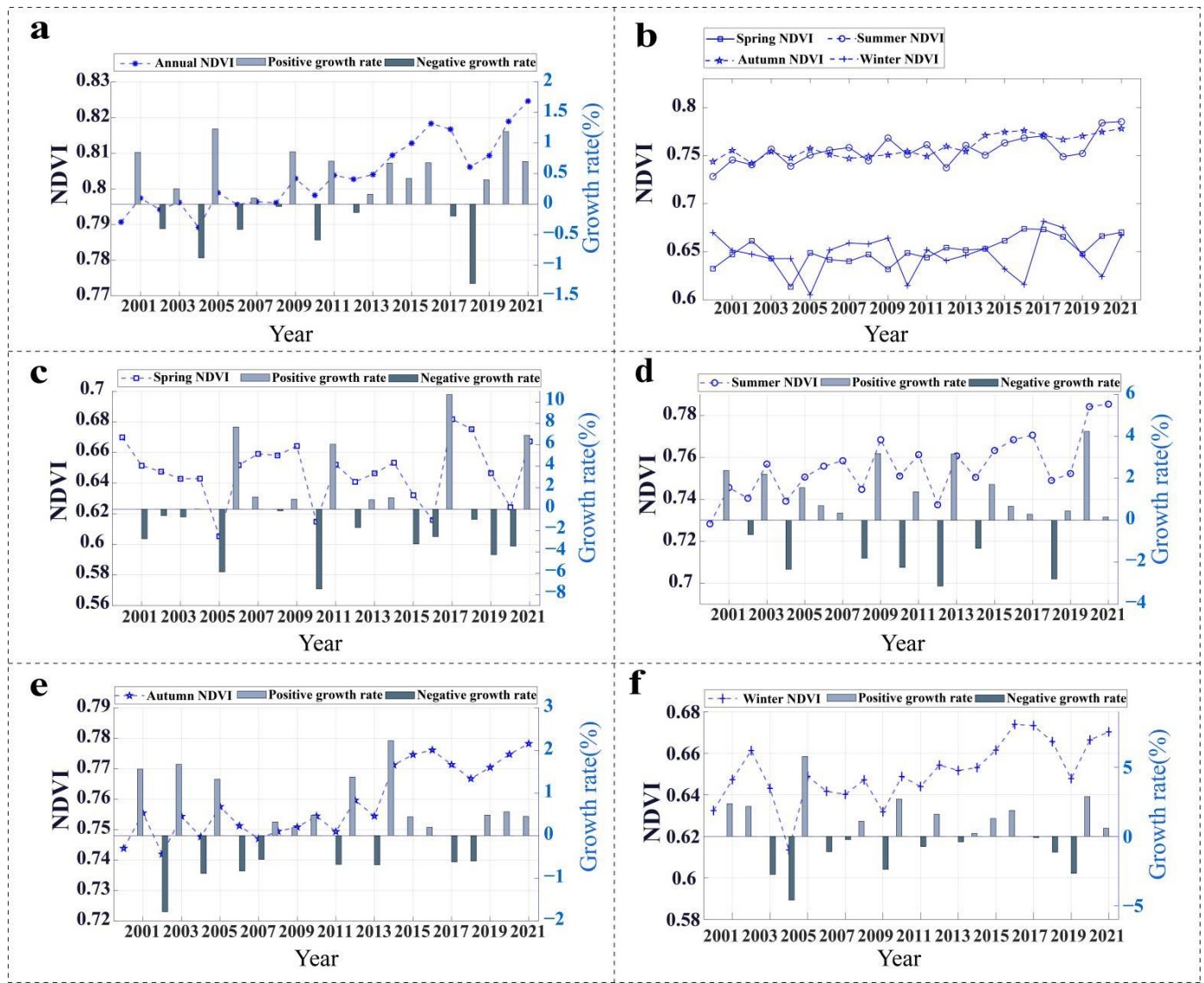

**Figure 3.** Temporal change of the NDVI from 2000 to 2021. (**a**) Average annual; (**b**) seasonal; (**c**) spring; (**d**) summer; (**e**) autumn; (**f**) winter.

### 3.1.2. Spatial Distribution Characteristics and Changes of the NDVI from 2000 to 2021

According to the NDVI remote sensing images of the study area from 2000 to 2021, the spatial distribution map of the multi-year average NDVI was obtained using the GIS raster calculator (Figure 4a). Then, according to the average distribution function [7], the vegetation cover was divided into five types: low vegetation coverage (0–0.2), relatively low vegetation coverage (0.2–0.4), medium vegetation coverage (0.4–0.6), relatively high vegetation coverage (0.6–0.8), and high vegetation coverage (0.8–1). According to the statistical results of classification and the spatial pattern of vegetation coverage (Figure 4), the NDVI distribution in the LMRB had apparent spatial heterogeneity, which was generally high in the south and low in the north, demonstrating a specific negative correlation with latitude; the lower the latitude, the higher the NDVI value. There was more than 95% vegetation coverage of the whole watershed greater than 0.6, and the area with vegetation coverage greater than 0.8 accounts for 61.44%. The watershed was dominated by high vegetation coverage and relatively high vegetation coverage.

Figure 4b,c shows the spatial distribution of NDVI in the study area in 2000 and 2021 and the proportion of NDVI at each level. The results demonstrated an increase in the high vegetation coverage from 58.14% in 2000 to 68.18% in 2021, indicating that the vegetation coverage in the study area was significantly improved in 2021 compared with that in 2000, especially in the cultivated land areas in southern Thailand and Cambodia, which were mainly converted from relatively high vegetation coverage to high vegetation coverage.

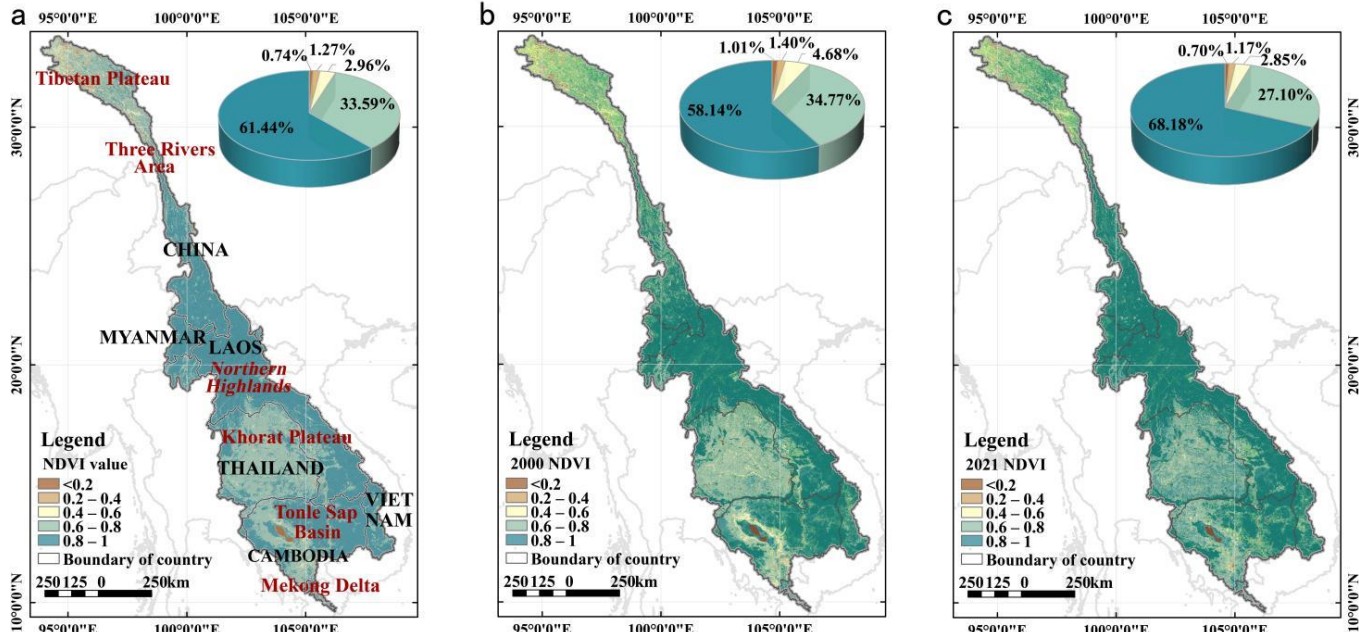

**Figure 4.** Spatial distribution of NDVI in the LMRB. (**a**) Average annual NDVI; (**b**) 2020 NDVI; (**c**) 2021 NDVI.

The spatial transition matrix of vegetation coverage in 2000 and 2021 was obtained (Table 4) using the Markov model of the IDRISI software, taking 2000 and 2021 as time nodes and according to the divided vegetation coverage types. This represents the interchange of NDVI space structures in 2000 and 2021. The results showed that from 2000 to 2021, the stability rates of low vegetation coverage, relatively low vegetation coverage, medium vegetation coverage, relatively high vegetation coverage, and high vegetation coverage in the LMRB were 49.33%, 37.69%, 26.75%, 45.91%, and 76.44%, respectively. Besides, the proportion of vegetation coverage improvement at each level was greater than degradation, and the improvement was mainly towards the adjacent level.

**Table 4.** NDVI transfer probability matrix of the LMRB in 2000 and 2021 (%).

| Year | Vegetation Type | 2021 | | | | |
|---|---|---|---|---|---|---|
| | | Low Vegetation Cover | Relatively Low Vegetation | Medium Vegetation Coverage | Relatively High Vegetation Coverage | High Vegetation Coverage |
| | Low vegetation cover | 49.33 | 36.80 | 7.40 | 4.80 | 1.67 |
| | Relatively low vegetation coverage | 6.13 | 37.69 | 34.75 | 16.58 | 4.85 |
| 2000 | Medium vegetation coverage | 0.30 | 3.89 | 26.75 | 58.83 | 10.23 |
| | Relatively high vegetation coverage | 0.30 | 0.17 | 2.65 | 45.91 | 51.23 |
| | High vegetation coverage | 0.04 | 0.41 | 0.41 | 22.70 | 76.44 |



*3.2. Trend and Mutation Analysis of NDVI from 2000 to 2021*

3.2.1. Sen-MK Trend Analysis

According to the Sen slope estimation and Mann–Kendall significance test, the 22 years of evolution trend and significance of each pixel were calculated (Figure 5a,b). The two results were superimposed and coupled into five levels using a GIS raster calculator (Figure 5c), thus obtaining the evolution characteristics of vegetation cover in the LMRB for the past 22 years. The results indicated that from 2000 to 2021, the area of the LMRB with an increasing vegetation coverage accounted for 66.59%, and the area with a decreasing vegetation coverage accounted for 18.88%. The increase region was much larger than the decrease region, and the overall change trend was positive, indicating that the vegetation in the basin was well protected. The significance test showed that 34.84% of the regions were significant, mainly those with improved NDVI. The area with significant increase in vegetation coverage accounts for 28.90% of the total watershed area, mainly distributed in Yunnan Province (forest land) in China and Thailand (cultivated land). The area of weak increase in vegetation coverage accounts for 37.69% of the total basin area, which is the largest proportion of trend types. The weak increase areas were evenly distributed in the basin countries, and the Qinghai Tibet Plateau of China and Laos were more distributed. The no-change area accounts for 14.53%, mainly distributed in the Qinghai Tibet Plateau of China and Laos. The area with a weak decrease in vegetation coverage accounts for 12.91%, mainly distributed in Laos, Cambodia, Vietnam, and the Qinghai Tibet Plateau. A significant decrease in vegetation coverage was the smallest trend type, accounting for 5.97%, mainly distributed in Cambodia in the south of the basin.

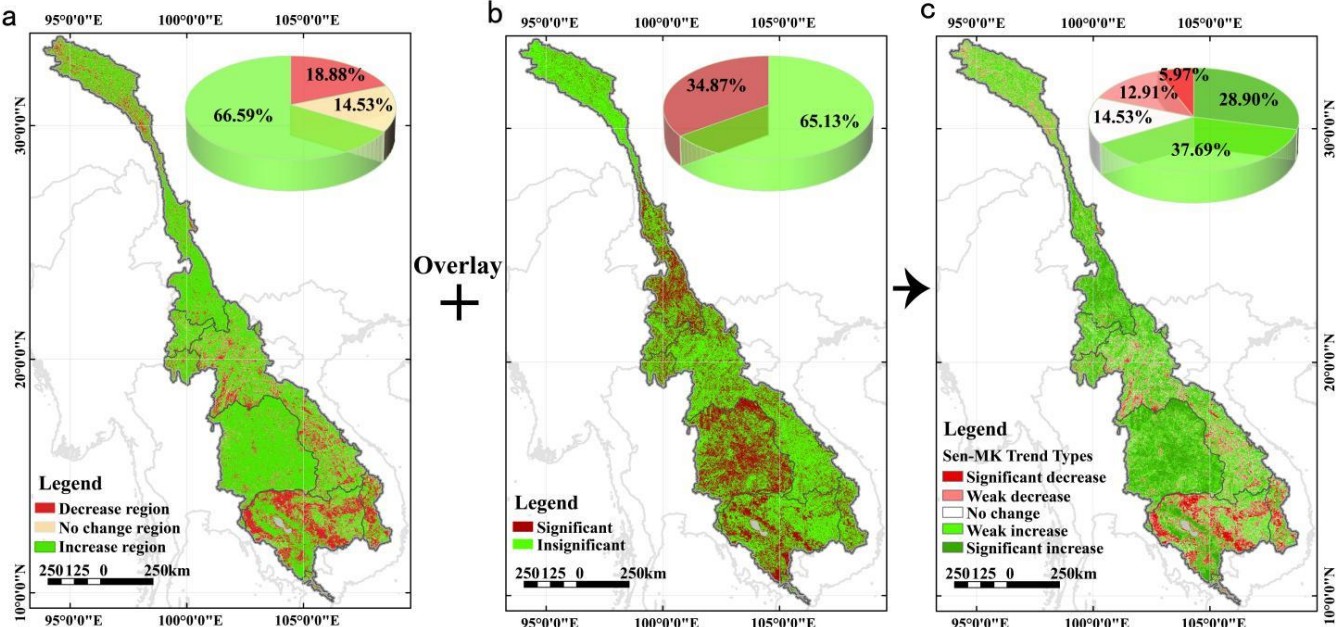

**Figure 5.** Spatial distribution of Sen-MK trend types. (**a**) Sen slope estimation results; (**b**) Mann–Kendall significance test; (**c**) Sen-MK trend coupling type.

3.2.2. Future Trends of NDVI

In order to reveal the trend and future persistence of NDVI changes, this study divided the Hurst exponent into three categories: persistence ($0.5 < H < 1$), anti-persistence ($0 < H < 0.5$), and uncertain ($H = 0.5$). The analysis indicates that the number of pixels with Hurst exponents between 0.5 and 1 accounts for 83.98%, and the NDVI in these regions will continue the evolution trend of the past 22 years. Besides, the proportion of pixels with Hurst exponents between 0 and 0.5 is 15.99%. The development trend of NDVI in these

areas is opposite to that of the past 22 years. The area with an H value of 0.5 accounts for 0.04%, and the future development trend of NDVI is unclear.

Further, the GIS raster calculator was utilized to superimpose the Hurst exponent reclassification result (Figure 6b) and the Sen-MK change trend result (Figure 6a) to obtain the coupling information of the changing trend with persistency (Figure 6c), including a total of 10 situation types. Then, they were classified into two development directions: benign and malignant. The results are shown in Figure 6 and Table 5. The following conclusions can be drawn from the analysis: (1) In the future, the NDVI in the study area will receive benign improvements, including four types: significant increase and persistence, weak increase and persistence, significant decrease and anti-persistence, and weak decrease and anti-persistence, accounting for 60.14% of the total area. It is mainly scattered in cultivated land, forest land, grassland, and other land types in the study area. From the perspective of countries, NDVI for Yunnan in China and Thailand will improve significantly. Human farming activities significantly affect these places, especially Thailand. Besides, it is found that among the vegetation that will improve in the future, about 2.71% of the areas were degraded in the past but had been improved with anti-persistence, and these areas are mainly scattered around the continuous improvement areas. (2) Compared with the past, the NDVI in the study area will be degraded and malignant in the future and also includes four types: significant decrease and persistence, weak decrease and persistence, significant increase and anti-persistence, and weak increase and anti-persistence, accounting for 25.29% of the total area of the whole region, mainly distributed in Cambodia, Laos, Vietnam, and the northern Qinghai Tibet Plateau. Among these areas, about 8.96% exhibited an improvement trend in the past 22 years. (3) Besides, about 14.53% of the region's NDVI development will remain unchanged in the future, and 0.04% of the region's NDVI development trend will be uncertain. These regions are mainly scattered in forests and grasslands. Particular attention should be paid to the regional vegetation that will be degraded in the future, especially cities, bare land, and plateau mountains, to prevent the ecological environment of the study area from deteriorating.

**Table 5.** Statistics of Sen-MK and Hurst coupling types.

| Development Direction | Hurst Sustainability | Sen MK Trend Type | Trend and Persistence Types | Number of Grids/Piece | Total/Piece | Proportion/% | Total Proportion/% |
|---|---|---|---|---|---|---|---|
| Benign | Persistence 0.5 < H < 1 | weak increase | weak increase and persistence | 3,659,827 | 7,456,410 | 29.52 | 60.14 |
| | | significant increase | significant increase and persistence | 3,460,548 | | 27.91 | |
| | anti-persistence 0 < H < 0.5 | significant decrease | significant decrease and anti-persistence | 22,285 | | 0.18 | |
| | | weak decrease | weak decrease and anti-persistence | 313,750 | | 2.53 | |
| Malignant | Persistence 0.5 < H < 1 | significant decrease | significant decrease and persistence | 735,276 | 3,135,665 | 5.93 | 25.29 |
| | | weak decrease | weak decrease and persistence | 1,290,168 | | 10.41 | |
| | anti-persistence 0 < H < 0.5 | weak increase | weak increase and anti-persistence | 978,299 | | 7.89 | |
| | | significant increase | significant increase and anti-persistence | 131,922 | | 1.06 | |
| Uncertain | Uncertain H = 0.5 | no change | uncertain | 4487 | 4487 | 0.04 | 0.04 |
| | | no change | no change | 1,801,105 | 1,801,105 | 14.53 | 14.53 |

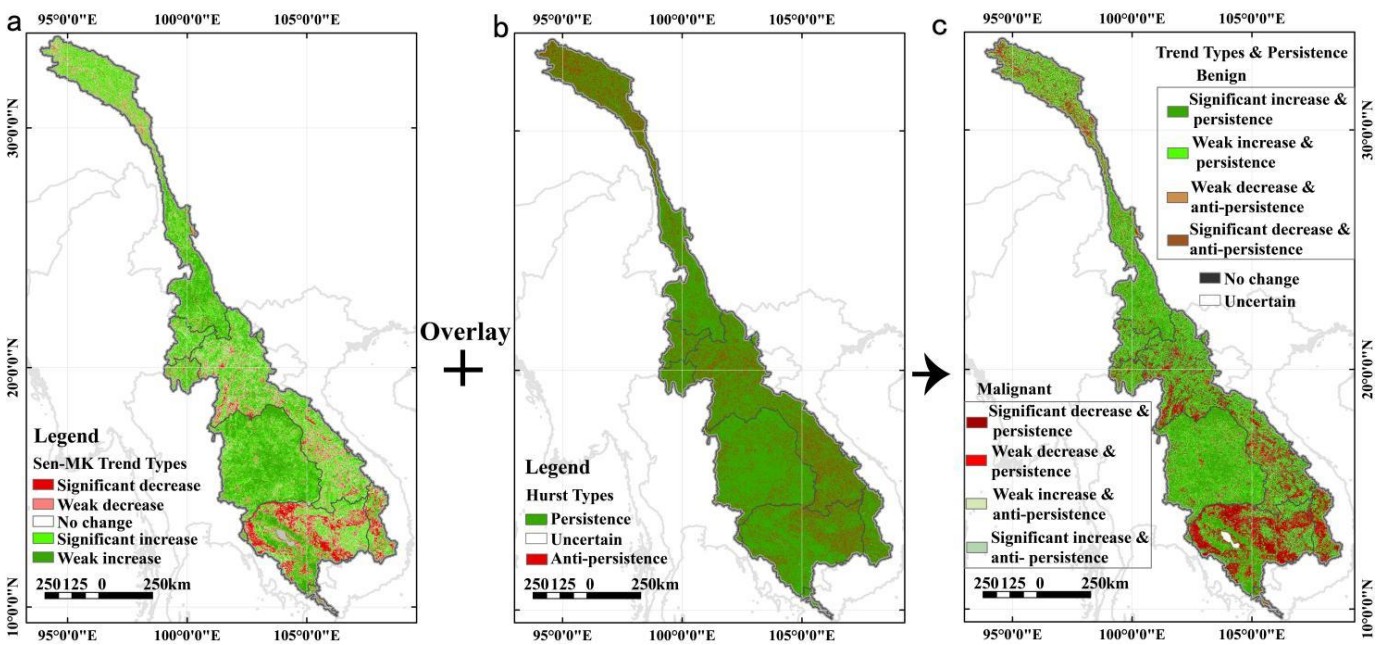

**Figure 6.** Spatial distribution of NDVI future persistence in the LMRB. (**a**) Sen-MK trend coupling type; (**b**) Hurst sustainability; (**c**) future trend type.

### 3.2.3. Nonlinear Mutation Analysis of NDVI

The BFAST01 algorithm was utilized to detect the NDVI mutation in the LMRB from 2000 to 2021 and obtain the spatial distribution map of the nonlinear mutation type and mutation year of NDVI (Figure 7). Figure 7 shows that the proportion of monotonic increase and decrease without breakpoints were the largest, accounting for 49.17% and 15.45%, respectively, followed by "interruption−+" 12.93%, "reversal−+" 8.76%, "interruption+−" 6.79%, "reversal+−" 6.30%, "monotonic increase+" 0.46%, and "monotonic decrease−" 0.23%. In addition to the monotonic increase and decrease types, nearly 35.38% of the NDVI areas were mutated. Therefore, when performing the trend analysis of NDVI, considering its nonlinear mutation, the trend can be evaluated more accurately. From the perspective of the overall changing trend, the NDVI trend mutation detected by BFAST01 can be classified into two types: the overall improvement area ("monotonic increase", "monotonic increase", interruption−+", "reversal−+") and the overall degradation area ("monotonic decrease", "monotonic decrease−","interruption+−", "reversal+−"), accounting for 71.24% and 28.76%, respectively. Their number was similar to this study's improvement and degradation trend of the Sen-MK trend classification. However, the BFAST01 detection result is more precise, while there are significant differences between different types. From the perspective of spatial distribution, the BFAST01 results were also similar to those of Sen-MK. The monotonic increase was the most critical trend type, widely distributed in countries in the study area. Since this type had not been interrupted, NDVI exhibited a long-term greening trend. The monotonic decrease was mainly distributed in Laos, Vietnam, and Cambodia in the lower reaches of the basin and the Qinghai Tibet Plateau in the upper reaches, while the vegetation coverage in these areas had not shown an improvement trend; "interruption increase type" ("monotonic increase+" and "interruption−+"); that is, the vegetation cover experienced some harmful disturbances, and short-term degradation appeared under the long-term gradual improvement, mainly scattered around the vegetation degradation area. The trend types of "interruption decrease type" ("monotonic decrease−" and "interruption+−") were rare, mainly scattered in the interior of the vegetation degradation area. Both "reversal+−" and "reversal−+" were inlaid around or inside the degraded areas, and the vegetation in these areas was greened or degraded due to some disturbances in the middle. It needs to be reminded that no matter the overall degradation or the improvement

area, the main focus should be on the areas where "greening" (improvement) reversals occur before and after the breakpoint for ecological protection and governance.

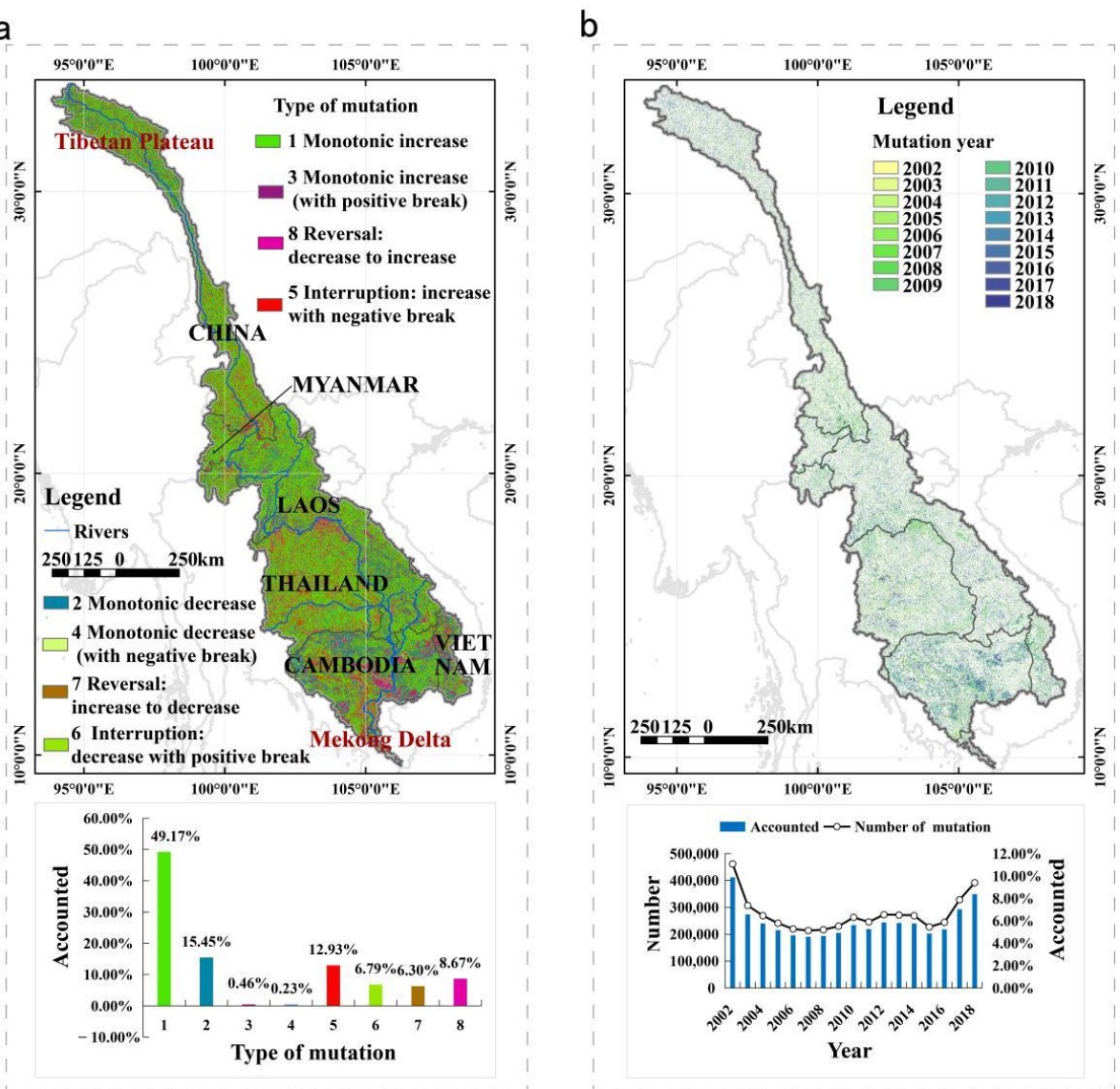

**Figure 7.** Mutation type (**a**) and mutation year (**b**) of BFAST01. The numbers 1,2,3,4,5,6,7,8 in Figure (**a**) represent the corresponding mutation type.

According to the significance test results, the proportion of significant trend type of BFAST01 was 43.49% (Figure 8), which is 8.62% more than that of the MK test. Specifically, the proportion of NDVI with both segments significant (or no break and significant) was 24.14%, mainly distributed in Yunnan in China and Thailand, while the number and distribution were similar to the MK test results. The case in which only the first and second segments are significant had ratios of 8.25% and 11.10%, respectively, mainly distributed in Yunnan in China, Thailand, and Cambodia. The pixels with both segments insignificant (or no break and not significant) account for 56.51%.

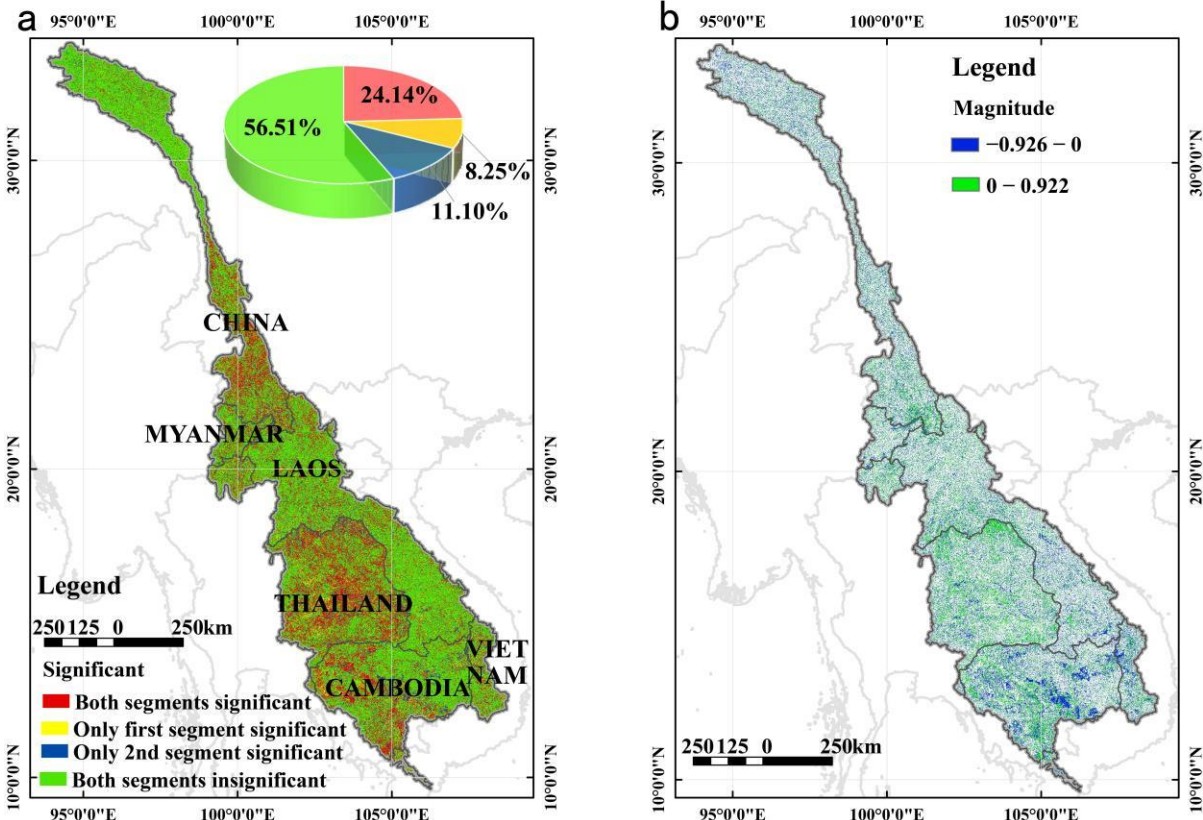

**Figure 8.** Significance test and change magnitude map of BFAST01 (**a**) Significance (**b**) magnitude.

In order to further analyze the breakpoint position of each trend's mutation type, the research also extracted the mutation year (i.e., trend interruption and transition) and change magnitude. As shown in Figure 8b, the disturbance years were mainly positive. The mutation year in this study refers to the trend in which breakpoints occurred in the trend type, while the monotonic increase and decrease types did not have breakpoints. Thus, these two types could not be observed in the mutation year. The proportion of the area where each mutation type was mutated in the mutation year is shown in Table 6. The statistical results show that the mutation years were mainly concentrated in 2002–2018. Among them, the year with the most breakpoints was 2002, with 461,812 breakpoints, followed by 2018 and 2017, and the number of breakpoints in other years was similar. Vertically, the breakpoint occurrence year was divided into five stages: 2002–2004, 2005–2007, 2008–2010, 2011–2013, and 2014–2018, while the proportion of mutations in each stage had a specific difference. Horizontal analysis indicated that the mutation years of "interruption−+" and "reversal−+" were mainly concentrated in 2002–2004, and they were the mutation types with the largest number of breakpoints, accounting for 36.54% and 24.51%, respectively, mainly in Thailand, Vietnam, and Cambodia. The mutation years of "interruption+−" and "reversal+−" were mainly concentrated in 2014–2018, indicating that the vegetation in the study area was mainly affected by negative disturbances in 2014–2018. The two stages of 2002–2004 and 2014–2018 had the most breakpoints among several stages, corresponding to the significant inflection points in 2004 and 2018 in the annual change trend of NDVI. Besides, the proportion of "monotonic increase+" and "monotonic decrease−" mutation in each stage is relatively uniform.

**Table 6.** Year distribution of changes in each mutation type.

| Type of Mutation | The Proportion of Area in the Year When Each Trend Changed (%) | | | | | Total |
|---|---|---|---|---|---|---|
| | 2002–2004 | 2005–2007 | 2008–2010 | 2011–2013 | 2014–2018 | |
| monotonic increase | - | - | - | - | - | - |
| monotonic decrease | - | - | - | - | - | - |
| monotonic increase+ | 0.29 | 0.19 | 0.24 | 0.19 | 0.40 | 1.31 |
| monotonic decrease− | 0.06 | 0.03 | 0.05 | 0.10 | 0.41 | 0.65 |
| interruption−+ | 11.57 | 6.57 | 4.79 | 3.75 | 9.86 | 36.54 |
| interruption+− | 1.59 | 2.04 | 3.37 | 5.19 | 6.99 | 19.19 |
| reversal+− | 1.99 | 1.64 | 2.53 | 3.91 | 7.74 | 17.80 |
| reversal−+ | 6.73 | 3.96 | 4.22 | 3.76 | 5.85 | 24.51 |
| Total | 22.21 | 14.44 | 15.19 | 16.91 | 31.25 | 100.00 |

Note: The meaning and full name of trend mutation types are shown in Table 2.

## 4. Discussion

### 4.1. Interannual Variation Analysis of NDVI

The LMRB is rich in forest resources and extensive agricultural planting area. The study shows that the NDVI of the basin shows a fluctuating growth trend as a whole from 2000 to 2021, indicating the development of the basin's overall vegetation towards a better trend, which is consistent with Han et al.'s studies [8] on the interannual change trend of vegetation in the Greater Mekong Subregion. However, according to the inter-annual trend change chart (Figure 3a), NDVI showed a relatively large downward trend in 2004 and 2018; that is, there was a sudden change, and the growth rate was negative. Many studies have shown that extreme climate events and the natural disasters caused by them are the main driving factors leading to changes in vegetation cover [1,29]. By reviewing the major events in the study area during the study period, we found that the decline of NDVI in 2004 may be due to the Indian Ocean tsunami in 2004 and the drought events in 2004–2005 shown in the Annual Mekong Flood Report 2017 [58]. Besides, the extreme flood disasters in 2017 also severely threatened the vegetation in the study area [29]. Moreover, the significant decline trend of NDVI in the study area in 2018–2019 may be caused by the extreme climate natural disasters in 2018, such as the catastrophic floods in southwestern India [59], the worst drought disaster in Thailand in nearly ten years in 2019, and the continuous global warming [60]. At the same time, the vegetation NDVI in the study area improved again after a short-term downward trend, which should be due to the corresponding measures taken by humans, such as China's artificial afforestation, the plan of returning farmland to forest, the protection of natural forests plans, and other ecological restoration projects and implementations of environmental policies. Besides, the social economic growth will also contribute to vegetation improvement [1,7,61]. Other studies show that strengthening agricultural activities and management levels can also improve the NDVI level [8], while the countries in the study area mainly have agricultural production, with a significant impact on human farming activities. Thus, the vegetation improvement was relatively apparent, especially in the farming areas in Thailand and Cambodia. In addition to the above-mentioned emergencies that may have some impact on vegetation coverage, when analyzing the factors affecting vegetation coverage, many studies generally discuss the impact factors of vegetation coverage from the perspectives of climate change and human activities [1,25,26,40]. Han et al. conducted an in-depth analysis of the influence of multiple factors such as climate change and human activities on the vegetation of the Greater Mekong Subregion in their research. The study believes that in terms of the impact type, degree, and scope, the human activities in the GMS deserve more attention than climate change [8]. We know that the Lancang-Mekong River Basin is rich in climate factors such as temperature, precipitation, and light, so it can be speculated that human activities may also be the main factor affecting the growth of NDVI. The line charts of seasonal changes in NDVI (Figure 3b–f) show that at the time node when the annual average NDVI declined, the seasonal NDVI also exhibited a corresponding downward trend, and the NDVI fluctuation

in each season was different. In spring, since the temperature rose and the vegetation began to germinate and grow, the NDVI fluctuation was significant. In winter, the vegetation was dormant, and the fluctuation was slighter than in spring. The NDVI values in summer and autumn were significantly higher than those in spring and winter, compatible with this region's tropical and subtropical monsoon climate characteristics with a high temperature and rainy summer and mild and less rainy winter. The sudden change in the annual change trend of NDVI in the study area confirms that the nonlinear mutation should be considered when detecting the trend of large-scale long-time series remote sensing NDVI [62]. This is also an essential part of the research and will be discussed.

### 4.2. Spatial-Temporal Pattern of NDVI

According to the spatial distribution of NDVI in the study area (Figure 4) and the statistical results, the vegetation coverage in the study area was excellent, which is attributed to the tropical and subtropical monsoon climate, good rainfall, and heat conditions. According to the NDVI classification spatial distribution map, the low vegetation coverage area accounts for a tiny proportion, mainly distributed in the northern Qinghai Tibet Plateau, as well as the lakes and waters in central Cambodia and the alluvial plains and estuary delta areas in the middle and lower reaches of the river. Combined with the altitude distribution of the study area, it can be seen that the vegetation coverage in the northern high-altitude area was lower than that in the southern part, indicating the significant impact of the terrain on vegetation coverage [22,63].

Besides, it was found that NDVI distribution is also closely related to land-use types [64]. Combined with the distribution of LULC in the study area (Figure 9), using GIS zoning statistics, we found that the land-use type with the highest vegetation coverage was forest land, which was 0.87, followed by grassland, cultivated land, construction land, and water area, while the LULC with the lowest vegetation coverage was ice and snow. Figure 9 shows that the high vegetation coverage was mainly distributed in the forest area, the LULC type in the relatively high vegetation coverage area was mainly cultivated land, and the LULC types in the low vegetation coverage area were mainly grassland, moss, shrubs, and bare lands. Combining the Sen-MK trend change type with land-use type, it was found that the vegetation in cultivated land areas had been significantly improved, demonstrating that human farming activities have promoted greening. It is worth reminding that, combined with the 2000 and 2020 Google Earth images shown in Figure 9, it can be seen that the overall land-use pattern in the study area has not changed significantly, and we can also see from Figure 4 that the overall spatial distribution of vegetation coverage in the LMRB has not changed much, so the area of vegetation coverage should not change much. Therefore, it can be judged that the Mekong Basin NDVI change is mainly that the total area of vegetation did not change significantly, but the vegetation status has become better [8,65]. However, vegetation coverage had a degradation trend to varying degrees in some urban areas, indicating that human activities such as economic development and urban expansion may also cause adverse interference with vegetation coverage [1]. The Hurst exponent results indicate that the proportion of areas where NDVI will be degraded will reach 25.29%. If appropriate protection and control measures are not taken for these areas where vegetation may deteriorate, the ecological environment of the study area may become worse. The good news is that, as shown in Table 7, most of these degraded areas began to grow after the breakpoint. Thus, these areas actually exhibited an improvement trend, which may be due to positive human activities that changed the vegetation degradation trend. Therefore, an accurate grasp of the mutation characteristics of NDVI trends is conducive to a more accurate analysis of the causes of vegetation degradation and the development of management programs.

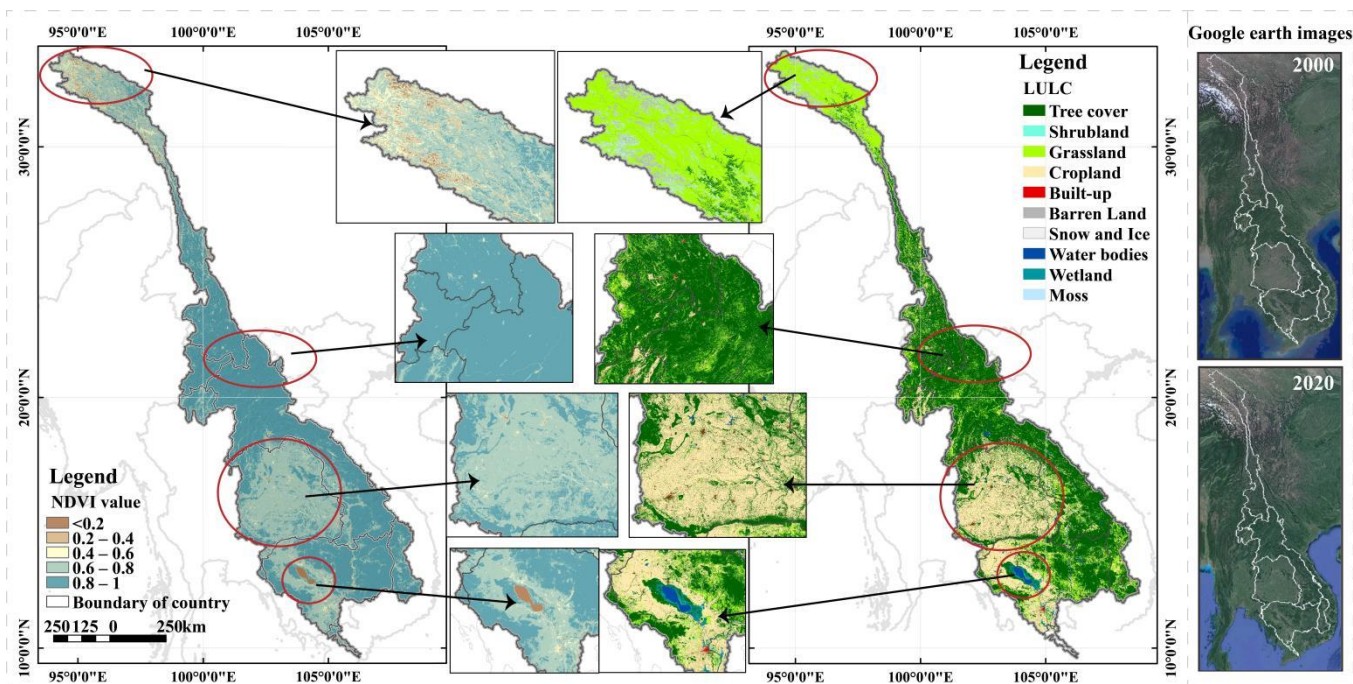

**Figure 9.** Distribution of NDVI and LULC in the study area.

**Table 7.** Conversion table of BFAST01 and Sen slope estimation results (%).

| BFAST01 \ Sen Trend | Monotonic Increase | Monotonic Decrease | Monotonic Increase+ | Monotonic Decrease− | Interruption−+ | Interruption+− | Reversal+− | Reversal−+ |
|---|---|---|---|---|---|---|---|---|
| Decrease | 1.30 | 55.06 | 0.00 | 1.21 | 8.34 | 12.92 | 8.49 | 12.68 |
| No change | 36.51 | 31.16 | 0.00 | 0.00 | 12.04 | 8.83 | 4.70 | 6.76 |
| Increase | 65.46 | 0.86 | 0.69 | 0.00 | 14.38 | 4.63 | 5.99 | 7.99 |

### 4.3. Sen-MK Trend and Nonlinear Mutation of BFAST01

This study adopted the BFAST01 method to determine the sudden events in the interannual trend of NDVI and the timing of the mutation, thus compensating for the deficiency of Sen slope estimation and the Mann–Kendall test that did not consider the mutation of long-term NDVI variability. Some studies have shown that the accuracy of the BFAST method in NDVI time series detection can reach 56% to 84% [66], and it can be applied to other disciplines dealing with seasonal or non-seasonal time series, such as hydrology, climatology, and econometrics [51]. Compared with other mutation detection methods, BFAST is less affected by seasonal differences and noise in time series and can detect changes faster [54]. Verbesselt et al. and Verbesselt et al. [52,67] presented a relatively complete description and introduction of this method. This study employed BFAST01 as the improved model of the BFAST method. When testing, we only focus on the maximum breakpoint in the NDVI interannual component and assume no seasonal change in the model. Many studies have verified the applicability of the BFAST01 model in vegetation NDVI [57], temperature change [68], and various fields [69,70].

The above discussion on the interannual variation of NDVI shows that NDVI is disturbed by various natural and human factors, so it is often not a simple linear change but may have complex fluctuations. Therefore, trend change detection should consider nonlinear mutations caused by the relevant disturbance events [55]. Table 7 compares the Sen slope estimation and BFAST01 trend classification results. It can be seen that the three types of increase, decrease, and no-change in the Sen slope estimation correspond to multiple trend mutation types of BFAST01. For example, 65.46% of the increase region in the Sen slope estimation corresponds to the monotonic increase type of BFAST01, and the rest corresponds to 0.69% of "monotonic increase+", 14.38% of "interruption−+", and 7.99% of

"reversal−+". Except for the areas in an improved state, there were a small number of areas in the improvement type estimated by Sen slope, which correspond to the degradation type in BFAST01, mainly "interruption+−" and "reversal+−", indicating that some of the improved NDVI were negatively interfered, and these disturbed areas were mainly scattered around the NDVI decrease area, of which the distribution in Cambodia was apparent, which requires particular attention for ecological governance. The comparison of the two methods shows that the traditional monotonous trend detection method may ignore some details in the long-term change trend of vegetation cover, while the BFAST01 method can just detect the possible interruption or conversion phenomenon due to various factors in the overall trend, identify different stages and their importance, and find areas that may be subject to vegetation degradation in the general trend and need special attention. Relevant research [6,23,62] has demonstrated the scientificity and applicability of the BFAST01 method. It is worth noting that this is not a denial of the Sen slope estimate, because the proportion of NDVI with the improved trend in BFAST01 mutation type (71.24%) was very close to that of the improved type (66.59%) in Sen slope estimation; both indicate that most of the vegetation cover in the study area was improved from 2000 to 2021. The conclusions of the two methods were compatible and similar to those of Fan et al. [65], Qiu et al. [71], and Han et al. [8] on the study of vegetation change in Lancang River, Lancang-Mekong River Basin, and Greater Mekong subregion, indicating the reliability of the results of this study. More detailed mutations of NDVI were identified in LMBR from 2000 to 2021 through BFAST01, which is the characteristic of this study. The two trend-analysis methods learn from each other to complement each other's strengths. Therefore, the combination of Sen-MK, Hurst exponent, and BFAST01 can express the history and future development dynamics of NDVI and present some details of nonlinear mutations in its historical evolution process, thus compensating for the shortcomings of using a specific method alone.

In order to verify the reliability of the BFSAT01 trend classification results, the study compared the images before and after the mutation year based on Google Earth time-series images to determine whether the mutation type detected by BFAST01 is correct. Part of the comparison results are shown in Figure 10. Taking the mutation year 2017 as an example, we show the Google Earth images before and after the mutation type ("monotonic decrease−") corresponding to the mutation year in Figure 10. It is found that the image change rule of the selected points conforms to the characteristics of the "monotonic decrease−" mutation type, that is, with 2017 as the negative interruption point, the vegetation coverage shows a downward trend from 2016 to 2018, and changes from cultivated land with vegetation to water. The changing rules of Google Earth images in other case sites are also consistent with the characteristics of the corresponding mutation type. It is proved again that the mutation detection of NDVI time series using BFAST01 is reliable, and the results are in line with the reality. In addition, BFAST01 is also an integration of Sen-MK results to some extent, which includes the trend types of Sen-MK in more detail. Therefore, the novelty of this study is that it employs a variety of statistical trend methods to express the details of NDVI trend changes and future sustainability, while taking into account both linear and nonlinear characteristics of long-term NDVI series in the study area.

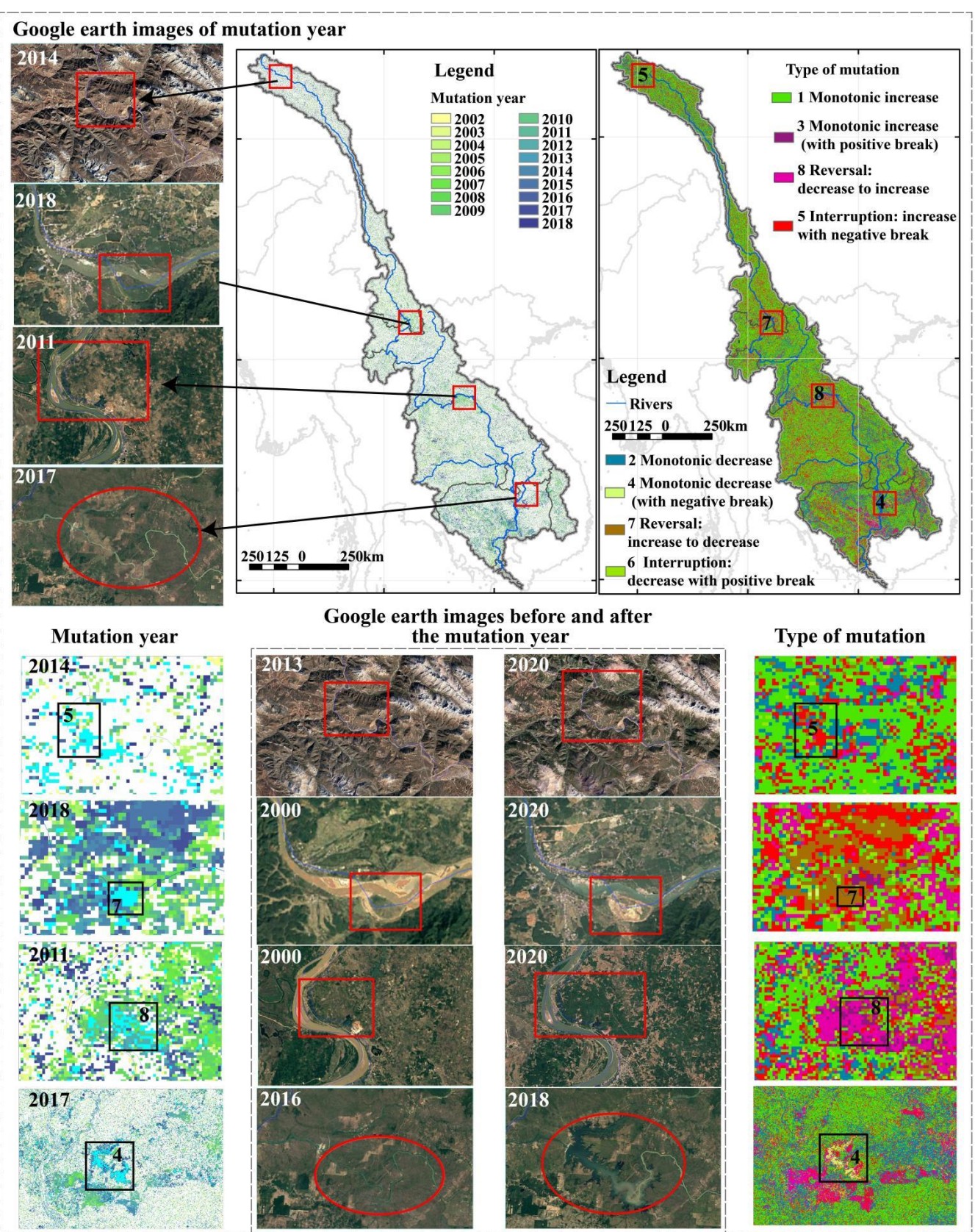

**Figure 10.** Visual comparison of Google Earth images before and after the mutation year. The numbers 4,5,7,8 in Figure represent the corresponding mutation type, in order to see the figure more clearly, use red squares and circles to highlight the range of the selected mutation year and mutation type in the picture.

*4.4. Limitations and Prospects*

Some limitations of this study include: (1) Due to data availability, the time scale is relatively short, affecting the accuracy of the research results [8,55]. Accordingly, GEE remote sensing big data processing cloud platform should be employed to conduct spatio-temporal fusion of multi-source data, thus overcoming the limitations of data time-scale and data accuracy. (2) The research innovatively combined Sen-MK, Hurst exponent, and BFAST01 to analyze the NDVI's trend evolution and mutation. Some studies have pointed out that the complexity and nonlinearity of vegetation cover may significantly affect the accuracy of predicting the future vegetation trend [7]. However, the existing studies using the Hurst exponent to predict the sustainability of NDVI do not consider the nonlinearity of vegetation cover. Therefore, this study will further explore the feasibility and scientificity of coupling analysis of the trend mutation nonlinearity detected by BFAST01 and Hurst sustainability to find a prediction method that can consider the nonlinear characteristics of NDVI and the future development direction. (3) Besides, the influencing factors of vegetation NDVI are complex, such as climate change, natural disasters, and other extreme events, as well as social and economic development, human activities, and other factors [1,7,8,72]. This study did not deeply analyze the driving factors of NDVI evolution due to space limitations. Thus, the next step will employ geographical detectors [16,73], correlation analysis [74], multiple linear regression analysis [8], and other methods to explore the influencing factors and reasons for NDVI changes in detail.

## 5. Conclusions

This study employed the MODIS NDVI data of the LMRB from 2000 to 2021 to complete the spatio-temporal pattern, trend change, mutation characteristics, and future sustainable status of NDVI in the study area in the last 22 years. The main conclusions are as follows:

(1) From 2000 to 2021, the NDVI of the LMRB generally showed a fluctuating upward trend, there was a significant mutation in 2004 and 2018, and the multi-year average NDVI value was 0.804. The NDVI distribution had apparent spatial heterogeneity, generally high in the south and low in the north, mainly with high and relatively high vegetation coverage. Although the vegetation coverage was excellent, the variation characteristics of different regions were different. (2) The Sen-MK trend shows that the area of the LMRB where NDVI improved and degraded in the last 22 years accounted for 66.59% and 18.88%, respectively. The improved area was much larger than the degraded area, indicating that the vegetation in the study area had been well protected. The Hurst exponent indicates that the proportions of areas where NDVI will be improved and degraded will be 60.14% and 25.29%, respectively, and special attention should be paid to areas where the future development trend will be degraded. (3) According to the BFAST01 nonlinear mutation test, the NDVI of the LMRB can be divided into eight mutation types in the last 22 years, and the proportion of the region with an increase in the mutation type is still larger than that with a decrease in the whole (71.24% > 28.76%). The mutation years occurred mainly from 2002 to 2018. The largest change percentage of mutation types during this period was "interruption−+", accounting for 36.54%, and the smallest percentage was "monotonic decrease−", accounting for only 0.65%. The trend results of NDVI detected by BFAST01 were very close to those of Sen-MK. Both indicate that the vegetation development prospects of the study area were good from 2000 to 2021. However, special attention should be paid to the areas where NDVI had abrupt changes and a decreasing trend. In the future, the natural and human-driven mechanisms of plant degradation will be further defined, and ecological protection suggestions will be put forward according to local conditions. The research has certain inspiration for the comprehensive analysis of long-term vegetation trends by using multiple statistical methods. In addition, the research results are conducive to the formulation of policies related to ecological environment protection and the promotion of regional sustainable development of the study area.

**Author Contributions:** Conceptualization, J.W. and X.Z.; methodology, X.Z. and J.L.; software, X.Z. and J.M.; validation, J.W. and X.Z.; formal analysis, J.L.; investigation, L.L. and J.M.; data curation, X.Z., J.Z. and L.L.; writing—original draft preparation, X.Z.; writing—review and editing, J.W. and J.L.; visualization, X.Z. and J.Z. All authors have read and agreed to the published version of the manuscript.

**Funding:** This work was supported by the Multi-government International Science and Technology Innovation Cooperation Key Project of National Key Research and Development Program of China (2018YFE0184300), the National Natural Science Foundation of China (41961060), the Tuojiang River Basin High-quality Development Research Center Program of China (TJGZL2022-15), and the Neijiang Normal University Program of China (2022YB17).

**Data Availability Statement:** All supporting data and access links used in the research were listed in Table 1.

**Acknowledgments:** We would like to express our respect and gratitude to the anonymous reviewers and editors for their professional comments and suggestions. In addition, we also want to thank the Program for Innovative Research Team (in Science and Technology) at the University of Yunnan Province (grant number IRTSTYN).

**Conflicts of Interest:** The authors declare no conflict of interest.

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
