# Peer review of "Linear and Nonlinear Characteristics of Long-Term NDVI Using Trend Analysis: A Case Study of Lancang-Mekong River Basin"

_remotesensing, doi:10.3390/rs14246271_

Round 1

Reviewer 1 Report

Although this is an interesting study, it could be useful for understanding the vegetation and ecology of the Mekong River basin. However, the article needs some revisions. Please address the comments to improve the quality of your article.

1. Please add references to the final paragraph of your introduction (lines 103 to 122). For example, lines 104-105, I recommend that the authors provide the following reference: Gao et al. (2022), Analyzing the critical locations in response of constructed and planned dams on the Mekong River Basin for environmental integrity, Environmental Research Communications, https://iopscience.iop.org/article/10.1088/2515-7620/ac9459. 

2. In section 2.1 (Study Area), authors may include an additional figure depicting river networks for the entire basin. Please see the aforementioned reference.

3. Please explain in details about Hurst Exponent in your section 2.3.2. Why it is important to address persistency of NDVI time series. Authors may follow the following reference for your convenience. Sarker et al. (2022), Spectral Properties of Water Hammer Wave, Appl. Mech. 2022, 3(3), 799-814; https://doi.org/10.3390/applmech3030047

4. Figures 2 and 3 require revision. Authors may use Python or Matlab to plot figures suitable for publication.

5. What makes this investigation novel? Possibly add more specifics to the manuscript, along with a discussion of the implications.

Author Response

Authors’ response to Reviewer 1 Comments

Remote Sensing Letters

Manuscript Number: remotesensing-2057520

Manuscript Title (Old): Spatial-temporal pattern and mutation analysis of NDVI in the Lancang-Mekong River Basin in the past 22 years

Manuscript Title (Revised): Linear and nonlinear characteristics of long-term NDVI using trend analysis: A case study of Lancang-Mekong River Basin

We are extremely grateful to you for reviewing our work and providing detailed comments and suggestions for revision, those comments are all valuable and very helpful for improving our manuscript. We really appreciate you giving us a chance to revise. We have studied comments carefully and have made correction one by one. A revision location is marked in the letter with a notation like (Revised edition: P1, Line 37-38). A copy of the fully revised manuscript that has all the changes highlighted in red color (Revised edition_highlight in red) is uploaded as an attachment. The responds are as flowing:

Specific comments:

Although this is an interesting study, it could be useful for understanding the vegetation and ecology of the Mekong River basin. However, the article needs some revisions. Please address the comments to improve the quality of your article.

Point 1: Please add references to the final paragraph of your introduction (lines 103 to 122). For example, lines 104-105, I recommend that the authors provide the following reference: Gao et al. (2022), Analyzing the critical locations in response of constructed and planned dams on the Mekong River Basin for environmental integrity, Environmental Research Communications, https://iopscience.iop.org/article/10.1088/2515-7620/ac9459.

Response: We would like to express our gratitude for your useful comments. We have revised the manuscript and added references to lines 108-129 of the manuscript according to your comments to enhance the persuasiveness.

Point 2: In section 2.1 (Study Area), authors may include an additional figure depicting river networks for the entire basin. Please see the aforementioned reference.

Response: Thank you for your advice! We have added the river network of the watershed in Figure 1 (line 149). Sorry to take your precious time to review it again.

Point 3: Please explain in details about Hurst Exponent in your section 2.3.2. Why it is important to address persistency of NDVI time series. Authors may follow the following reference for your convenience. Sarker et al. (2022), Spectral Properties of Water Hammer Wave, Appl. Mech. 2022, 3(3), 799-814; https://doi.org/10.3390/applmech3030047

Response: Thank you for the valuable comments. We have added the explanation of Hurst exponent in the article according to the references you provided (Revised edition: P6, Line 223-230). The Hurst exponent represents structure over asymptotically longer periods. It is a term used to describe long-range dependence, which quantifies relative tendency of time series to regress strongly to the mean or cluster in a certain direction. It is a measure of a data series’ ‘mild’ or ‘wild’ randomness. It has to do with time series autocorrelations and the pace at which they drop as the lag between pairs of values grows longer. It was established in hydrology for the purpose of calculating the optimum dam size. NDVI is a typical remote sensing time series data, and its value changes with time with strong correlation, in recent years, it has been widely utilized in future sustainability studies of vegetation cover changes, we believe that it is scientific to use Hurst exponent to describe the long-term dependence of NDVI. Appreciate for the comment. I hope this response is clear enough to meet the requirements. If there is still something unsuitable, please feel free to enlighten us, I hope you can give me the opportunity to correct it, and we will cherish it.

Point 4: Figures 2 and 3 require revision. Authors may use Python or Matlab to plot figures suitable for publication.

Response: Thanks for your suggestion. We have redrawn Figures 2 and 3 using Matlab (line 264 and 317), them are looks like much better than the old one.  

Point 5: What makes this investigation novel? Possibly add more specifics to the manuscript, along with a discussion of the implications.

Response: Thank you very much for your good questions and valuable suggestions! We have added some specifics in the manuscript to express the innovation of the study, and added a paragraph in the discussion section of 4.3 to further elaborate the reliability and particularity of the research (Revised edition: P20, Line 643-660). In addition, also introduce the special and novelties of this study in the last paragraph of the Introduction (P3, Line 108-129). First of all, the selection of the research area has an international perspective. The Lancang-Mekong River Basin spans 6 countries in total and is a hot spot for many studies in South Asia and Southeast Asia; Secondly, the research methods and ideas are novel. This study uses traditional long-term series trend analysis methods such as Sen slope estimation, Manna-Kendall significance test, and Hurst exponent to analyze the historical trend and future sustainability of NDVI, and uses the BFAST01 model to detect NDVI mutation. It is the first time that linear and nonlinear trend analysis methods are combined to study the trend of long-term remote sensing data, which is a special feature of the study. We also introduced the principle of the BFAST01 model in detail in section 2.3.3 of the research method, and further discussed the advantages of BFAST01 in section 4.3, as well as the comparison of the joint use of several methods. Thanks for your comments again. We hope our response can be satisfied with you. If there is anything inappropriate, please point it out again, and we will definitely revise it seriously.

All in all, thank you very much for your considering our manuscript and giving us  opportunity to revise for potential publication in Remote Sensing.

Best regards,

The first author:Xuzhen Zhong

Email address1: zxzxuzhen@njtc.edu.cn   

Email address2: 904213389@qq.com  

Corrsponding author: Jinliang Wang

Email address: jlwang@ynnu.edu.cn

Reviewer 2 Report

In this study, a time-series analysis framework is presented to analysis changes of a region using remote sensing products. Some statistical approaches are employed to obtain results. Please consider following comments to improve the quality of the manuscript:

1-      Title: it seems to me that the use of some statistical methods and comparison of their results are the originality of the manuscript. So it would be interesting to mention it in the title. In this form, it seems a case study paper is presented.

2-      In the introduction, statistical methods and related papers are presented but it is necessary to mention new recent publications. Please consider the following papers:

Zhang, Yang. "A time-series approach to detect urbanized areas using biophysical indicators and landsat satellite imagery." IEEE Journal of Selected Topics in Applied Earth Observations and Remote Sensing 14 (2021): 9210-9222.

Yan, Kai, et al. "Performance stability of the MODIS and VIIRS LAI algorithms inferred from analysis of long time series of products." Remote Sensing of Environment 260 (2021): 112438.

Tehrani, Nadia Abbaszadeh, et al. "Time-series analysis of COVID-19 in Iran: A remote sensing perspective." COVID-19 pandemic, geospatial information, and community resilience. CRC Press, 2021. 277-290.

Martínez, Beatriz, et al. "Exploring Ecosystem Functioning in Spain with Gross and Net Primary Production Time Series." Remote Sensing 14.6 (2022): 1310.

Qu, Ge, et al. "Characterization of Long-Time Series Variation of Glacial Lakes in Southwestern Tibet: A Case Study in the Nyalam County." Remote Sensing 14.19 (2022): 4688.

3-      The maximum value composite (MVC) method was then utilized to attain the annual maximum NDVI dataset for dynamic vegetation analyses, Why? The statistical methods can support all times and it is not clear why this processing is necessary.

4-      I did not see any preprocessing steps applied in NDVI products. It must be mentioned in the text, why the preprocessing steps are not necessary.

5-      Table 1: please use English words in the manuscript.

6-      Manna-Kendall or Mann-Kendall?

7-      Equations are not visually interesting, please use an appropriate font.

8-      It is necessary to present a part in the methodology regarding validations of results. I cannot understand how we can accept results.

9-      Is there any way to integrate results of the statistical methods regarding vegetation changes?

10-   Please compare results of the best method (at least visually) with google earth time series images.

Author Response

Authors’ response to Reviewer 2 Comments

Remote Sensing Letters

Manuscript Number: remotesensing-2057520

Manuscript Title (Old): Spatial-temporal pattern and mutation analysis of NDVI in the Lancang-Mekong River Basin in the past 22 years

Manuscript Title (Revised): Linear and nonlinear characteristics of long-term NDVI using trend analysis: A case study of Lancang-Mekong River Basin

We are extremely grateful to you for reviewing our work and providing detailed comments and suggestions for revision, those comments are all valuable and very helpful for improving our manuscript. We really appreciate you giving us a chance to revise. We have studied comments carefully and have made correction one by one. A revision location is marked in the letter with a notation like (Revised edition: P1, Line 37-38). A copy of the fully revised manuscript that has all the changes highlighted in red color (Revised edition_highlight in red) is uploaded as an attachment. The responds are as flowing:

Specific comments:

In this study, a time-series analysis framework is presented to analysis changes of a region using remote sensing products. Some statistical approaches are employed to obtain results. Please consider following comments to improve the quality of the manuscript:

Point 1: Title: it seems to me that the use of some statistical methods and comparison of their results are the originality of the manuscript. So it would be interesting to mention it in the title. In this form, it seems a case study paper is presented.

Response: We would like to express our gratitude for your useful comments. We have revised the title of the manuscript according to your comments. The new title is: “Linear and nonlinear characteristics of long-term NDVI using trend analysis: A case study of Lancang-Mekong River Basin”.We hope our modifications will satisfy your suggestions.

Point 2: In the introduction, statistical methods and related papers are presented but it is necessary to mention new recent publications. Please consider the following papers:

Zhang, Yang. "A time-series approach to detect urbanized areas using biophysical indicators and landsat satellite imagery." IEEE Journal of Selected Topics in Applied Earth Observations and Remote Sensing 14 (2021): 9210-9222. 

Yan, Kai, et al. "Performance stability of the MODIS and VIIRS LAI algorithms inferred from analysis of long time series of products." Remote Sensing of Environment 260 (2021): 112438. 

Tehrani, Nadia Abbaszadeh, et al. "Time-series analysis of COVID-19 in Iran: A remote sensing perspective." COVID-19 pandemic, geospatial information, and community resilience. CRC Press, 2021. 277-290. 

Martínez, Beatriz, et al. "Exploring Ecosystem Functioning in Spain with Gross and Net Primary Production Time Series." Remote Sensing 14.6 (2022): 1310.

Qu, Ge, et al. "Characterization of Long-Time Series Variation of Glacial Lakes in Southwestern Tibet: A Case Study in the Nyalam County." Remote Sensing 14.19 (2022): 4688.

Response: Many thanks to you for providing so many useful documents. We have inquired and carefully read the literature on statistical methods recommended by you. These articles more comprehensively express the application of methods such as Theil-Sen slope and Mann-Kendall test in urbanized area, GPP, NPP, LAI, Glacial Lake, and COVID-19, etc. We have cited and introduced these articles in the manuscript (Revised edition: P2, Line 87-93), thank you again and hope that our changes meet your requirements.

Point 3: The maximum value composite (MVC) method was then utilized to attain the annual maximum NDVI dataset for dynamic vegetation analyses, Why? The statistical methods can support all times and it is not clear why this processing is necessary.

Response: Thank you for the valuable comments. We have added the explanation of MVC in the article according to your suggestions (Revised edition: P4, Line 167-175). The MVC algorithm was proposed by Holben in the early days for the production of time series products. It selects the maximum vegetation index value in the synthesis cycle as the composite value, which can remove most of the cloud and atmosphere effects. Its removal effect on cloud and bidirectional reflectance noise is obvious, with high computing efficiency, which is conducive to the production of vegetation index products, can provide users with important reference values of NDVI [36,37], and is one of the most widely used algorithms in NDVI data processing [38]. Many studies have used this algorithm [39-40], and some studies have mentioned that MVC images are highly correlated with the dynamics of green vegetation [41]. The overall vegetation coverage in the Lancang-Mekong River Basin is good, and the NDVI differences in various regions are not very large, so MVC method can better show the NDVI differences. Therefore, this paper chooses this algorithm for NDVI composite. Appreciate for the comment. I hope this response is clear enough to meet the requirements. 

Point 4: I did not see any preprocessing steps applied in NDVI products. It must be mentioned in the text, why the preprocessing steps are not necessary.

Response: Thanks for your suggestion. We added some steps of NDVI data preprocessing in section 2.2 (Revised edition: P4-5, Line 165-178). We used the GEE platform to download MODIS NDVI data, and performed preprocessing on the GEE platform, such as splicing, cropping, and projection. In addition, the MVC method was used to eliminate the interference of the atmosphere, cloud, scanning angle and solar zenith angle on the digital value of the image. Thanks again for taking time out of your busy schedule to review it.

Point 5: Table 1: please use English words in the manuscript.

Response: Sorry, reviewers, it was our carelessness that did not check that there is no English expression here. We have corrected in the manuscript (Revised edition: P5, Table 1). Thanks again for your reminder. We must be more careful in the future and not make low-level mistakes. If there is still a problem here, please point it out to me again, we must seriously modify it.

Point 6: Manna-Kendall or Mann-Kendall?

Response: We would like to express our gratitude to you for pointing out some spelling errors in our manuscript. It was our carelessness. We have checked and proofread the entire article for extual errors. We found that there are 7 misspellings of Mann-Kendall in the full text. Respectively in the Abstract (Revised edition: P1, Line 20), Introduction (Revised edition: P3, Line 122), Methods (Revised edition: P5, Line 191 and 195), section 3.2.1 ï¼ˆRevised edition: P12, Line 356 and 379) and section 4.3 ï¼ˆRevised edition: P19, Line 594). We have corrected all these spelling mistakes. A copy of the fully revised manuscript is uploaded as an attachment. We hope our modifications can be satisfied with you, thank you again.

Point 7: Equations are not visually interesting, please use an appropriate font.

Response: We greatly appreciate issues pointed out and suggestions provided by the you. In the manuscript, we used the Math Type to write equations. According to your suggestion, we uniformly adjusted the fonts of the formulas to Times New Roman. If the modified form is not beautiful enough, I hope you can point it out again, and we will modify it again strictly according to the standard.

Point 8: It is necessary to present a part in the methodology regarding validations of results. I cannot understand how we can accept results.

Response: We have to say that your comments have played a great role in improving the quality of our manuscript. Thank you very much for your valuable comments, which have inspired us a lot. According to your this suggestion and the 10th suggestion, we mentioned the idea of the result verification in the method of the manuscript (Revised edition: P20, Line 643-660), and made a comparison chart to add the verification analysis of the research results in the discussion section (Revised edition: P21, Figure 10). The study used MODIS data for NDVI time series analysis, and the reliability of the results should be verified by relevant methods. For remote sensing image analysis, using another image to compare results is a method often used in many articles. And the google earth time series images you proposed are the approved satellite imagery for verification, so we adopted it.

Point 9: Is there any way to integrate results of the statistical methods regarding vegetation changes?

Response: Thank you reviewer, according to your suggestion, we have done more thinking. we think that the NDVI change results obtained by Sen-MK and BFAST01 statistics can be integrated with each other to some extent. The research analysis shows that the changing trend of NDVI obtained by the two methods is consistent. On the one hand, Sen-MK trend classification results only have five types: significant increase, significant decrease, weak increase, weak decrease and no change, while the NDVI types detected by BFAST01 include monotonic increase, monotonic decrease, "monotonic increase+", "monotonic decrease -", "interruption-+", "interruption+-", "reversal+-", "reversal-+" 8 types in total. Not only can Sen MK trend classification be further subdivided into BFAST01 trend types, but also BFAST01 trend types can be integrated into fewer types. However, the purpose of our research is to express the future sustainability of NDVI through Sen-MK and Hurst index, and express NDVI trend types in more detail through BFAST01. Therefore, several methods were jointly used in this article. We also made some supplementary explanations in the manuscript (Revised edition: P20, Line 654-657). Because of limited knowledge and understanding, I don't know whether the answer meets your requirements. If there is anything inappropriate, please feel free to enlighten us and give us a chance to correct it. We will definitely revise it again. Thank you again sincerely.

Point 10: Please compare results of the best method (at least visually) with google earth time series images.

Response: Thank you for your advice! We have used google earth time series images in the manuscript to compare the results (Revised edition: P20, Line 43-660), sorry for wasting your precious time to review my paper again. If there is still something unsuitable, I hope you can give me the opportunity to correct it, and we will cherish it.

All in all, thank you very much for your considering our manuscript and giving us  opportunity to revise for potential publication in Remote Sensing.

Best regards,

The first author:Xuzhen Zhong

Email address1: zxzxuzhen@njtc.edu.cn   

Email address2: 904213389@qq.com  

Corrsponding author: Jinliang Wang

Email address: jlwang@ynnu.edu.cn

Reviewer 3 Report

As a transboundary river in Asia, the land cover change in the Mekong River Basin has always been a research hotspot. This study took the Lancang-Mekong River Basin as the study area, and employs the Sen slope estimation, Manna-Kendall test, and Hurst exponent based on the MODIS NDVI data from 2000 to 2021 to investigate the spatial and temporal evolution trend and future sustainability of its NDVI. I personally believe this is an interesting study, which will receive concerns from international readers about the vegetation changes in the Mekong River basin. However, regarding the current version, the study results should be interpreted in details. The significance behind the NDVI changes should also be dug deeper. My major concerns are as follows:

1. The authors have reported that over the past 22 years, NDVI has displayed very marked changes in the Mekong Basin. Is it possible to analyze the causes of this change? Climate change or human activities, which one is the major contributor?

2.There may be two scenarios for the changes in NDVI: one is that the total area of vegetation did not change significantly, but the vegetation status has become better; the other is that the total area of vegetation has increased significantly. In the Mekong Basin, which scenario dominated the changes? Ay reasons?

3. Any implications of the study findings suggested for policy, practice, theory, and subsequent research?

Author Response

Authors’ response to Reviewer 3 Comments

Remote Sensing Letters

Manuscript Number: remotesensing-2057520

Manuscript Title (Old): Spatial-temporal pattern and mutation analysis of NDVI in the Lancang-Mekong River Basin in the past 22 years

Manuscript Title (Revised): Linear and nonlinear characteristics of long-term NDVI using trend analysis: A case study of Lancang-Mekong River Basin

We are extremely grateful to you for reviewing our work and providing detailed comments and suggestions for revision, those comments are all valuable and very helpful for improving our manuscript. We really appreciate you giving us a chance to revise. We have studied comments carefully and have made correction one by one. A revision location is marked in the letter with a notation like (Revised edition: P1, Line 37-38). A copy of the fully revised manuscript that has all the changes highlighted in red color(Revised edition_highlight in red) is uploaded as an attachment. The responds are as flowing:

Specific comments:

As a transboundary river in Asia, the land cover change in the Mekong River Basin has always been a research hotspot. This study took the Lancang-Mekong River Basin as the study area, and employs the Sen slope estimation, Manna-Kendall test, and Hurst exponent based on the MODIS NDVI data from 2000 to 2021 to investigate the spatial and temporal evolution trend and future sustainability of its NDVI. I personally believe this is an interesting study, which will receive concerns from international readers about the vegetation changes in the Mekong River basin. However, regarding the current version, the study results should be interpreted in details. The significance behind the NDVI changes should also be dug deeper. My major concerns are as follows:

Point 1: The authors have reported that over the past 22 years, NDVI has displayed very marked changes in the Mekong Basin. Is it possible to analyze the causes of this change? Climate change or human activities, which one is the major contributor?

Response: Thank you for the valuable comments and we extend our sincerest gratitude to you. We analyzed the influencing factors of some special cases of NDVI changes in the Discussion Section 4.1. In fact, except for some extreme climates or emergencies that may affect vegetation coverage, when analyzing the factors affecting vegetation coverage, we should conduct a comprehensive discussion from the perspectives of climate change and human activities. According to your suggestion, we found a study similar to the study area. In his study, Han et al.'s conducted an in-depth analysis of the impact of multiple factors such as climate change and human activities on the vegetation of the Greater Mekong Subregion. The study believes that in terms of the impact type, degree and scope, the human activities in the GMS deserve more attention than climate change. We know that the Lancang-Mekong River Basin is rich in climate factors such as temperature, precipitation, and light, so it can be speculated that human activities may also be the main factor affecting the growth of NDVI. Due to space limitations, this study did not quantitatively analyze the reasons for the changes in NDVI. Thus, the next step will employ geographical detectors, correlation analysis , multiple linear regression analysis , and other methods to explore the influencing factors and reasons for NDVI changes in detail. Referring to the article by Han et al.'s et al., we also added an analysis of the reasons for the overall change of NDVI in Section 4.1 of the manuscript (Revised edition: P17, Line 524-535), hoping to improve the persuasiveness of the article. Sorry for wasting your precious time to review my paper again, if there is still something unsuitable, please feel free to enlighten us, I hope you can give me the opportunity to correct it, and we will cherish it.

Point 2: There may be two scenarios for the changes in NDVI: one is that the total area of vegetation did not change significantly, but the vegetation status has become better; the other is that the total area of vegetation has increased significantly. In the Mekong Basin, which scenario dominated the changes? Any reasons?

Response: We would like to express our gratitude for your useful comments. We have made more analysis on the change scenarios and reasons of NDVI in Section 4.2 of the manuscript (Revised edition: P18, Line 570-577). From the NDVI spatial distribution map (Figure 4) of the study area, we can see that the overall spatial distribution of vegetation coverage in the Lancang-Mekong River Basin has not changed much in the past 22 years, but the overall vegetation status has improved. Our research mainly uses a variety of statistical trend analysis methods to analyze the NDVI changes of each pixel in the past 22 years. Therefore, the conclusion we get is also for the trend of the pixel itself. Referring to related studies [8, 65], combined with google earth time series images, and the actual situation of the study area, the overall situation of vegetation change in the study area should be dominated by the first scenario you mentioned. The LULC in the study area is mainly dominated by forest land and cultivated land. For the Mekong Basin area, it is mainly cultivated land, that is, there are rich farming activities. From the google earth images, it can be seen that the overall land use pattern in the study area has not changed significantly, so the area of vegetation coverage should not change much. Therefore, strengthening farming activities is conducive to improving vegetation conditions [8]. We hope our modifications can be satisfied with you, thank you again.

Point 3: Any implications of the study findings suggested for policy, practice, theory, and subsequent research?

Response: Many thanks for your comments, it inspired us to think more. The research has certain inspiration for the comprehensive analysis of long-term vegetation trends by using multiple statistical methods, and can provide certain references for related follow-up studies, so it has some theoretical significance. In addition, the research results are conducive to the formulation of policies related to ecological environment protection and the promotion of regional sustainable development of the study area. We have appropriately included a statement of the implications of the study findings in our conclusions (Revised edition: P22, Line 715-719). Thanks again for taking time out of your busy schedule to review it. If there is anything inappropriate, please point it out again, and we will definitely revise it seriously.

All in all, thank you very much for your considering our manuscript and giving us  opportunity to revise for potential publication in Remote Sensing.

Best regards,

The first author:Xuzhen Zhong

Email address1: zxzxuzhen@njtc.edu.cn   

Email address2: 904213389@qq.com  

Corrsponding author: Jinliang Wang

Email address: jlwang@ynnu.edu.cn

Round 2

Reviewer 1 Report

Thanks for the revision. 

Reviewer 2 Report

-

Reviewer 3 Report

The revised version is satisfactory; It is fine for the publication in Remote Sensing. However, at the end of the paper, the reference format still has some small problems. Please update the reference list before accepting for publication.